# FlipNet: Fourier Lipschitz Smooth Policy Network for Reinforcement Learning

## Abstract

Deep reinforcement learning (RL) is an effective method for decision-making and control tasks. However, RL-trained policies encounter the action fluctuation problem, where consecutive actions significantly differ despite minor variations in adjacent states. This problem results in actuators' wear, safety risk, and performance reduction in real-world applications. To address the problem, we identify the two fundamental reasons causing action fluctuation, i.e. policy non-smoothness and observation noise, then propose the **F**ourier **Lip**schitz Smooth Policy **Net**work (**FlipNet**). FlipNet adopts two innovative techniques to tackle the two reasons in a decoupled manner. Firstly, we prove the Jacobian norm is an approximation of Lipschitz constant and introduce a Jacobian regularization technique to enhance the smoothness of policy network. Secondly, we introduce a Fourier filter layer to deal with observation noise. The filter layer includes a trainable filter matrix that can automatically extract important observation frequencies and suppress noise frequencies. FlipNet can be seamlessly integrated into most existing RL algorithms as an actor network. Simulated and real-world experiments show that FlipeNet has excellent action smoothness and noise robustness, achieving a new state-of-the-art performance. The code and videos are publicly available [1].

## 1 Introduction

Deep reinforcement learning (RL) has become a powerful approach for addressing optimal control tasks in physical environments (Guan et al., 2021; Li, 2023). Neural networks, capable of modeling complex nonlinear functions (Hornik et al., 1989; Kidger & Lyons, 2020), are commonly used as the container for the control policy fitted by RL. However, RL-trained policies often encounter the action fluctuation problem, where consecutive actions exhibit significant variations despite minor differences in the adjacent observations. While this problem is often overlooked during simulation and training stages, it will result in serious issues in real-world application like performance reduction, actuators' wear, and safety risk (Song et al., 2023). This problem is prevalent in various scenarios, including drone control (Mysore et al., 2021; Shi et al., 2019), robot arm manipulation (Yu et al., 2021), and autonomous driving (Cai et al., 2020; Chen et al., 2021; Wasala et al., 2020).

In order to make RL more applicable in real-world scenarios, researchers are working hard to solve the problem. CAPS (Mysore et al., 2021) and L2C2 (Kobayashi, 2022) introduce penalty terms in actor loss, which indicate the action similarity in successive time steps or the action similarity under close states. SR²L (Shen et al., 2020; Zhao et al., 2022) trains policy network using adversarial noise, which maximizes the action difference under actual state and adversarial state. PIC (Chen et al., 2021) and TAAC (Yu et al., 2021) design two-stage policies by using one network to output the current action, and the other to output action inertia scalar or make choice between the current and the last action. MLP-SN (Takase et al., 2020) and LipsNet (Song et al., 2023) smoothes control action by constraining the Lipschitz constant of policy network. However, CAPS and L2C2 suffer from sensitive hyperparameter tuning, and their sampling of close states complicate RL algorithms. SR²L, PIC, and TAAC need special policy evaluation or policy improvement mechanisms, which means they cannot be used in traditional RL algorithms. MLP-SN suffers from several performance loss and LipsNet is limited to the network with piecewise linear activation functions. Furthermore,

---

[1]Project page: https://iclr-anonymous-2025.github.io/FlipNet

none of them have directly dealt with the observation noise. There is still an open challenge to smooth control action in a way that is effective, simple, and applicable across various RL algorithms.

In this paper, we propose a novel policy network structure, named **F**ourier **Lip**schitz Smooth Policy **Net**work (**FlipNet**), achieving action smoothing in RL effectively, simply and flexibly. We identify the fundamental reasons for causing action fluctuation are the non-smoothness of policy network and the existence of observation noise. FlipNet adopts two corresponding techniques to directly tackle them. Firstly, we propose Jacobian regularization technique to constrain the Lipschitz constant of policy network. We prove the Jacobian norm is an approximation of neural network's Lipschitz constant, thereby enhancing the smoothness of policy function fitted in the policy network by regularizing the Jacobian norm. Secondly, we propose a Fourier filter layer to filter observation noise. In this layer, fast Fourier transform (FFT) is used to obtain

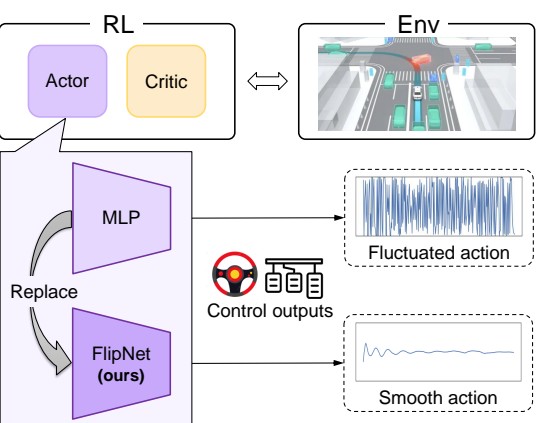

Figure 1: **FlipNet outputs smooth action.**

the frequency features of sequential observations, and a trainable filter matrix is used to automatically extract important frequencies in observation and suppress noise frequencies. Finally, we package FlipNet as an user-friendly PyTorch module. FlipNet has three superiorities compared to previous works: (1) FlipNet directly tackles the two fundamental reasons causing action fluctuation, while previous works do not consider them at the same time; (2) The user-friendly packaging of FlipNet does not disturb original RL algorithm, allowing application in various RL algorithms, including TRPO (Schulman et al., 2015), TD3 (Fujimoto et al., 2018), and DSAC (Duan et al., 2021), etc.; (3) FlipNet has better overall performance, including the control performance and action smoothness.

**Experiment results.** Simulated and physical experiments verify that FlipNet has achieved the state-of-the-art (SOTA) performance. For the simulated tasks, we conduct experiments on the double integrator environment and DeepMind control suite benchmark (DMControl). For example, in DM-Control's walker environment, FlipNet increases the total average return (TAR) by 3.4% and reduces the action fluctuation ratio (AFR) by 35.5% compared to LipsNet, which is the previous SOTA network. Additionally, an experiment of physical vehicle robot is implemented to test on real-world application, where the vehicle robot is going to track given trajectories and avoid moving obstacle under various noise levels. Results show that FlipNet increases the average TAR by 5.8% and reduces the average AFR by 90.0% compared to the multilayer perceptron (MLP).

**Technical contributions.** FlipNet is a novel network, addressing the action fluctuation problem in the real-world applications of RL. Our contributions are four-fold: **(1)** We identify the two fundamental reasons that cause action fluctuation, and propose a policy network named FlipNet to tackle the two reasons in a decoupled manner; **(2)** We demonstrate that the Jacobian norm serves as an approximation of Lipschitz constant, and propose a Jacobian regularization technique to enhance the smoothness of policy network; **(3)** We propose a trainable Fourier filter layer, capable of automatically extracting valuable observation frequencies while suppressing noise frequencies; **(4)** We conduct extensive experiments on both simulated and real-world tasks to validate FlipNet's SOTA performance. The code is publicly released to facilitate the implementation and future research.

## 2 PRELIMINARIES

### 2.1 ACTOR-CRITIC REINFORCEMENT LEARNING

Actor-critic method, consisting of an actor network and a critic network as shown in Figure 1, forms the backbone of many RL algorithms. The actor network fits a policy $\pi : \mathcal{S} \rightarrow \mathcal{A}$ that mapping from state space to action space. Therefore, the actor network is also called as policy network. The goal of RL is to train a policy $\pi$ maximizing the expected return: $J_\pi = \mathbb{E}_{\tau \sim \rho_\pi} \left[ \sum_{t=0}^{T} \gamma^t r_t \right]$, where

$\rho_\pi$ is the distribution of state-action trajectory induced by policy $\pi$, $T$ is the termination time of an episode, $0 \le \gamma \le 1$ is the discount factor, and $r_t$ represents the reward. The critic network fits a value function $V(s)$ or $Q(s, a)$, mapping from the state-action pairs to the expected returns, to evaluate the actions taken by the actor.

In policy evaluation phase, the critic is updated by minimizing the temporal difference (TD) error. For example, the Q-value network in DDPG (Lillicrap et al., 2015) parameterized by $\varphi$ is updated by

$$\min_\varphi \left( \mathbb{E}_{s,a,r,s' \sim \mathcal{D}} \left[ Q_\varphi(s, a) - r - \gamma Q_{\varphi_{\text{targ}}}(s', a') \right] \right)^2, \tag{1}$$

where $\mathcal{D}$ is the replay buffer, $s'$ is the next state, $a'$ is the next action obtained by the target actor network, and $Q_{\varphi_{\text{targ}}}$ is the return estimated by the target critic network.

In policy improvement phase, the actor is updated by maximizing the expected return predicted by the critic. Taking DDPG as an example again, the actor network is updated by minimizing the actor loss function:

$$\mathcal{L} = \mathbb{E}_{s \sim \mathcal{D}} \left[ -Q_\varphi(s, \pi(s)) \right]. \tag{2}$$

## 2.2 ACTION FLUCTUATION RATIO

Action fluctuation ratio (AFR) is an index to quantitatively measure the fluctuation level of control action (Chen et al., 2021; Song et al., 2023). It is defined as

$$\xi(\pi) = \mathbb{E}_{\tau \sim \rho_\pi} \left[ \frac{1}{T} \sum_{t=1}^{T} ||a_t - a_{t-1}|| \right], \tag{3}$$

where $\rho_\pi$ is the distribution of state-action trajectory induced by policy $\pi$, $T$ is the termination time of episodes, $a_t$ and $a_{t-1}$ are two adjacent actions, and $|| \cdot ||$ is the norm of action difference vector [2].

Beside the total average return (TAR), AFR is also an important indicator to evaluate policies' performance in the real world. The smaller AFR is, the smoother action sequence policy $\pi$ has.

## 3 METHODOLOGY

### 3.1 REASONS IDENTIFICATION OF ACTION FLUCTUATION

To ensure that RL agents produce smooth actions, it is necessary to first identify the root cause of action fluctuation. In decison-making and control tasks, the actions are calculated by the policy network $\pi$ according to the current observation $o_t$, i.e. $a_t = \pi(o_t)$. And the current observation $o_t$ is composed by the current state $s_t$ and observation noise $\xi_t$, i.e. $o_t = s_t + \xi_t$. The rate of action change over time is $\frac{\mathrm{d}a_t}{\mathrm{d}t} = \frac{\mathrm{d}\pi(o_t)}{\mathrm{d}o_t} \cdot \frac{\mathrm{d}o_t}{\mathrm{d}t}$, then we can derive that

$$\left\| \frac{\mathrm{d}a_t}{\mathrm{d}t} \right\| \le \left\| \frac{\mathrm{d}\pi(o_t)}{\mathrm{d}o_t} \right\| \cdot \left( \left\| \frac{\mathrm{d}s_t}{\mathrm{d}t} \right\| + \left\| \frac{\mathrm{d}\xi_t}{\mathrm{d}t} \right\| \right). \tag{4}$$

To mitigate action fluctuation, $\|\frac{\mathrm{d}a_t}{\mathrm{d}t}\|$ must be controlled within a reasonable range. From Equation (4), we know $\|\frac{\mathrm{d}a_t}{\mathrm{d}t}\|$ is affected by three parts: a red term of policy derivative, reflecting the level of policy smoothness; a blue term of noise change rate, reflecting the level of observation noise; and an inherent derivative term of the target dynamics system.

Based on the above analysis, the two fundamental reasons that causes action fluctuation can be identified: (1) the non-smoothness of policy network, and (2) the existence of observation noise.

**Non-smoothness of policy network.** A non-smooth policy network means that RL fits a non-smooth policy function mapping from the state to control action. The mapping function has significant output differences even the inputs are closely adjacent. Consequently, when the state changes

---

[2]Throughout the paper, $|| \cdot ||$ denotes the 2-norm of a vector or a matrix.

with time, a non-smooth action sequence is generated. Appendix C visualizes the effect of a non-smooth policy.

**Existence of observation noise.** The noise results in the discontinuous changes in observations, making the actions produced by the policy network at the adjacent time stamps erratically differ. Even if the policy function fitted by the policy network is smooth enough, actions can still be fluctuated because of the erratic observation noise.

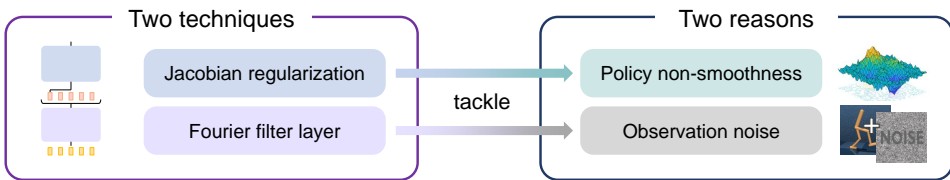

Figure 2: **The proposed two techniques address the two fundamental reasons** that cause action fluctuation in a straightforward, direct, and decoupled manner.

Therefore, the control action won't be smooth enough unless the two fundamental reasons are both under control. Previous works do not identify the two reasons clearly, and none of them consider the two aspects at the same time. Although some works recognize the effect of observation noise, they choose to improve the robustness by reducing the Lipschitz constant of policy network (Takase et al., 2020; Song et al., 2023), i.e. enhancing the smoothness of policy network, rather than directly filtering observation noise. Such a non-decoupled approach results in actions being insufficiently smooth, and a loss of performance when sufficient action smoothness is required. In this paper, we propose the Jacobian regularization technique and the Fourier filter layer to respectively tackle the two fundamental reasons in a straightforward, direct, and decoupled manner, as shown in Figure 2.

## 3.2 JACOBIAN REGULARIZATION

**Definition 3.1** (Local Lipschitz Constant). *Suppose $f : \mathbb{R}^n \to \mathbb{R}^m$ is a continuous neural network. The $K(x)$ is defined as the local Lipschitz constant of $f$ on the neighborhood of $x$:*

$$K(x) = \max_{x_1, x_2 \in \mathcal{B}(x,\rho)} \frac{\|f(x_1) - f(x_2)\|}{\|x_1 - x_2\|}, \tag{5}$$

*where $\mathcal{B}(x, \rho)$ denotes the open ball area with radius $\rho > 0$ centered at the point $x$ in the Euclidean space, i.e. $\mathcal{B}(x, \rho) = \{x' : \|x' - x\| < \rho\}$.*

Lipschitz constant characterizes the landscape smoothness of a function. By viewing the policy network as a mapping function, Lipschitz constant could reflect the smoothness of the policy function fitted by RL. A lower Lipschitz constant means a smoother policy function, which leads to smoother actions (Ames et al., 2016; Kobayashi, 2022; Song et al., 2023; Takase et al., 2020). MLP-SN (Takase et al., 2020) constrains the Lipschitz constant by applying spectral normalization (SN) (Miyato et al., 2018) on each layer of policy network. However, applying SN usually leads to severe performance loss, because the desired network-wise Lipschitz continuity is realized by layer-wise Lipschitz constraints (Bhaskara et al., 2022; Wu et al., 2021). LipsNet (Song et al., 2023) proposes a network-wise method, Multi-dimensional Gradient Normalization (MGN), to constrain the Lipschitz constant. However, MGN needs to set an initial Lipschitz constant manually, which may damage RL's exploration ability. And MGN needs to calculate Jacobian matrix during forward propagation, which makes the policy network not applicable in high real-time tasks.

To overcome the above challenges, we propose the Jacobian regularization method to conveniently reduce the Lipschitz constant of policy network. The Jacobian norm is commonly used as an index of function's smoothness and robustness (Hoffman et al., 2019; Lee et al., 2023). In Theorem 3.1, we prove that Jacobian norm is an approximation of the local Lipschitz constant.

**Theorem 3.1** (Lipschitz's Jacobian Approximation). *For a continuously differential neural network $f : \mathbb{R}^n \to \mathbb{R}^m$, the Jacobian norm $\|\nabla_x f\|$ is an approximation of the local Lipschitz constant of $f$ on the infinitesimal neighborhood of $x$, i.e. $K(x) \approx \|\nabla_x f\|$.*

*Proof.* See Appendix A in the supplementary material. □

According to Theorem 3.1, the Jacobian norm is an approximation of local Lipschitz constant and we know that Lipschitz constant reflects function's landscape smoothness, therefore we can conveniently enhance the policy smoothness by reducing Jacobian norm. The tailored actor loss becomes

$$\mathcal{L}' = \mathcal{L} + \lambda_k \left\| \nabla f \right\|, \tag{6}$$

where $\mathcal{L}$ is the original actor loss, and $\lambda_k$ is a coefficient. The proposed Jacobian regularization is superior to the Lipschitz constraint methods used in MLP-SN and LipsNet because: (1) Jacobian regularization is a network-wise rather than layer-wise constraint method, avoiding severe performance loss; (2) Jacobian regularization does not need to set a initial Lipschitz constant manually, not damaging the exploration ability of RL; (3) Jacobian regularization dose not need to calculate Jacobian matrix during forward propagation, applicable in high real-time tasks.

### 3.3 FOURIER FILTER LAYER

Fourier Transform is a widely used frequency analysis tool, which can also be employed in neural networks for feature extraction (Lee-Thorp et al., 2022; Rao et al., 2021). To mitigate the action fluctuation caused by observation noise, we propose the Fourier filtering layer based on Fourier Transform. The workflow of Fourier filter layer is shown in Figure 3.

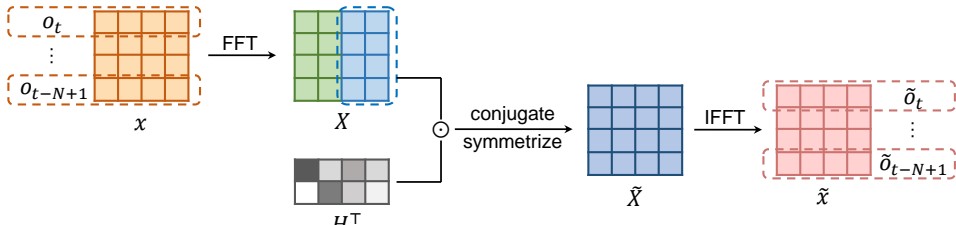

Figure 3: **Workflow of Fourier filter layer.** Firstly, FFT converts historical observations to frequency feature matrix $X$. Then, half of $X$ is multiplied by a trainable filter matrix $H$, and a complete matrix $\tilde{X}$ is generated by conjugate symmetrizing. Finally, IFFT converts $\tilde{X}$ to filtered time-domain signals.

Given $N$ historical observations $o_t, o_{t-1}, \cdots, o_{t-N+1} \in \mathbb{R}^D$ where $D$ denotes the dimension of features, the Fourier filter layer concatenates them as a matrix $x \in \mathbb{R}^{N \times D}$, and calculates the frequency feature matrix $X \in \mathbb{C}^{N \times D}$ using 2D discrete Fourier transformation:

$$X_{u,v} = \sum_{n=0}^{N-1} \sum_{d=0}^{D-1} x_{n,d} \cdot e^{-j2\pi \left( \frac{un}{N} + \frac{vd}{D} \right)}, \tag{7}$$

where $x_{n,d}$ denotes the $d$-th feature of the $n$-th observation signal, $X_{u,v}$ denotes the element located at the $u$-th row and $v$-th column of the frequency feature matrix $X$, and $j$ represents the imaginary unit. When the length of historical observations is less than $N$, the missing parts are padded with 0. In FFT, Zero-padding does not alter the primary frequency components of the signal, and it merely increases the spectral resolution (Jung et al., 2019). The magnitude of $X_{u,v}$ denotes the signal intensity at the frequency combination $(u, v)$, where $u$ and $v$ are frequency indices rather than the actual frequency values. Since the observations only consist of real values, the resulting matrix $X$ exhibits conjugate symmetry, i.e. $\overline{X_{u,v}} = X_{N-u,D-v}$. It means that half of $X$ could represent the complete information contained in the signal.

Then, half of $X$, denoted as $X_{\text{half}} \in \mathbb{C}^{N \times \lfloor \frac{D}{2} \rfloor + 1}$, is subjected to a Hadamard product with a trainable filter matrix $H \in \mathbb{C}^{N \times \lfloor \frac{D}{2} \rfloor + 1}$. After that, a complete matrix $\tilde{X} \in \mathbb{C}^{N \times D}$ is restored by conjugate symmetrizing the product matrix:

$$\tilde{X} = \text{symmetrize}(X_{\text{half}} \odot H). \tag{8}$$

By choosing $H$ as a complex matrix instead of real matrix, the Fourier filtering layer can not only alter frequency amplitudes but also perform feature extraction. The magnitudes of the elements in

$H$ determine which frequency is suppressed or strengthened. To enable the noise filtering capability of policy network, we encourage the magnitudes of elements in $H$ to be as small as possible. In this way, policy network can automatically extract valuable frequencies and filter out less relevant frequencies where noise may exist. Consequently, the actor loss is tailored from $\mathcal{L}'$ in Equation (6) into

$$\mathcal{L}'' = \mathcal{L} + \lambda_k \left\| \nabla f \right\| + \lambda_h \left\| H \right\|_F, \tag{9}$$

where $\left\| H \right\|_F$ is the Frobenius norm of $H$, and $\lambda_h$ is a coefficient.

Finally, the resulted frequency feature matrix $\tilde{X}$ is recovered to the time-domain signals by 2D inverse discrete Fourier transformation:

$$\tilde{x}_{n,d} = \frac{1}{ND} \sum_{u=0}^{N-1} \sum_{v=0}^{D-1} \tilde{X}_{u,v} \cdot e^{j2\pi\left(\frac{un}{N} + \frac{vd}{D}\right)}. \tag{10}$$

Because $\tilde{X}$ is a conjugate symmetric matrix, the matrix $\tilde{x} \in \mathbb{R}^{N \times D}$ becomes a real matrix. By slicing rows from the matrix $\tilde{x}$, the filtered features $\tilde{o}_t, \tilde{o}_{t-1}, \cdots, \tilde{o}_{t-N+1} \in \mathbb{R}^D$ are obtained. The signal $\tilde{o}_t$, representing the filtered feature corresponding to the current timestamp, is selected as the input for subsequent layers. The subsequent layers form a subnetwork $f$, which is processed by Jacobian regularization for a low Lipschitz constant. The overall structure of FlipNet is shown in Figure 4. The pseudocode of FlipNet is illustrated in Appendix B.

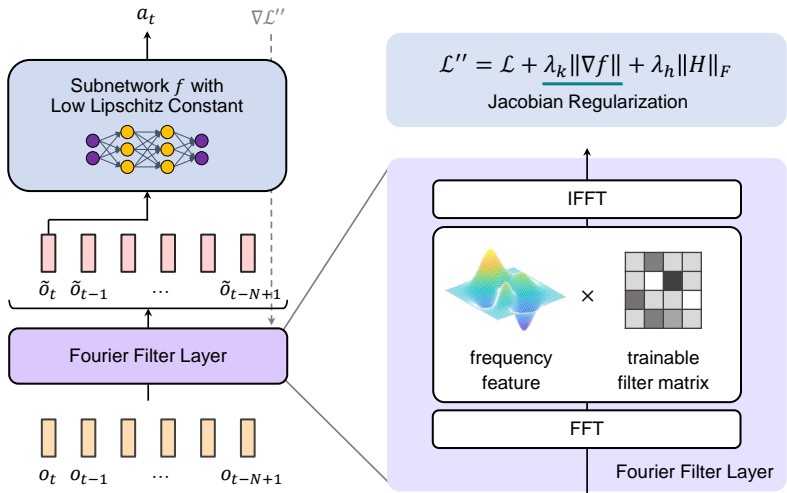

Figure 4: **Overall structure of FlipNet.** Historical observations are processed by Fourier filter layer, where a trainable filter matrix is used for frequency selection. The filtered feature $\tilde{o}_t$ is inputted into a subnetwork whose Lipschitz constant is constrained by Jacobian regularization. The parameters in FlipNet are updated by tailored actor loss $\mathcal{L}''$.

### 3.4 USER-FRIENDLY PACKAGING

To facilitate research and usage, we package FlipNet as an user-friendly PyTorch (Paszke et al., 2019) module. A backward hook function is used in the module. When network's backward propagation is called, the hook function will awake to automatically replace the gradient derived from $\mathcal{L}$ by the gradient derived from $\mathcal{L}''$. In this way, users don't need to redefine the actor loss and any other elements in RL, making FlipNet applicable in almost all

```
net = FlipNet()
out = net(input)
...
loss.backward()
```

Figure 5: **User-friendly deployment.**

actor-critic RL algorithms like DDPG (Lillicrap et al., 2015), TD3 (Fujimoto et al., 2018), PPO (Schulman et al., 2017), TRPO (Schulman et al., 2015), SAC (Haarnoja et al., 2018) and DSAC (Duan et al., 2021), etc. As shown on the right, practitioners can use FlipNet just like using an MLP. The code is publicly available at [3].

---

[3]Project page: https://iclr-anonymous-2025.github.io/FlipNet

# 4 EXPERIMENTS

## 4.1 DOUBLE INTEGRATOR

Double integrator is a classic linear quadratic control task, which is commonly used to test the performance of controllers. In the environment, a particle is moving along an axis without resistance (Song et al., 2023). The observations include position $x$ and velocity $v$ of the particle. The control action is particle acceleration $a$ that parallel to the axis. A schematic diagram of the environment is shown in Figure 6.

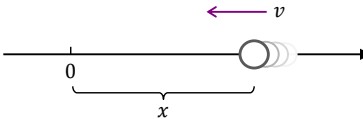

The reward function is $r = -2x^2 - v^2 - a^2$, which incentives the particle to remain stable at the origin, i.e. $x = 0, v = 0, a = 0$. The Infinite-time Approximate Dynamic Programming (INFADP) (Li, 2023), a model-based RL algorithm, is used for train without noise. When testing policy networks, the particle has nonzero initial position and velocity, and the noise for each observation dimension is distributed in $U(-0.2, 0.2)$. More details and hyperparameters are shown in Appendix D.

Figure 6: **Double integrator.**

The results are presented in Figure 7 and 8. In Figure 7(a), 30 episodes are simulated starting from the same initial state. The solid line and shadow area respectively denote the mean and standard deviation of actions. The shadow areas imply the action fluctuation amplitude of FlipNet is much smaller than that of MLP, and is on par with LipsNet. Figure 7(b) depicts action trajectories for a single episode, which reveals that FlipNet has better action continuity than LipsNet under the same level of action fluctuation amplitude. This conclusion is confirmed again by Figure 7(c), where action trajectories are decomposed by FFT and the action frequency induced by FlipNet is shown to be more distributed in the low-frequency range.

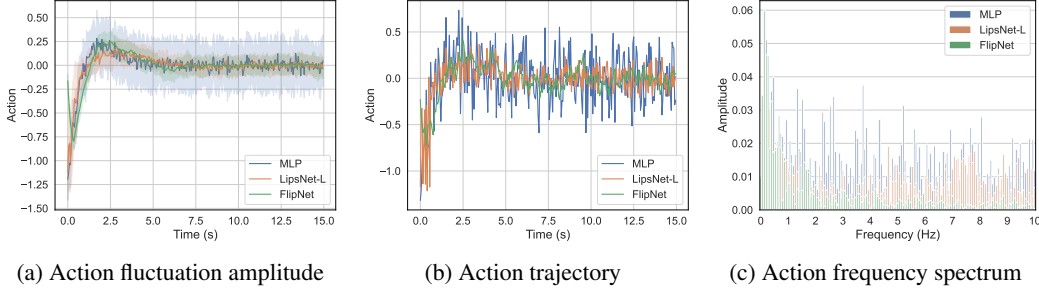

(a) Action fluctuation amplitude     (b) Action trajectory     (c) Action frequency spectrum

Figure 7: **Action in double integrator environment.** (a) The action fluctuation amplitude of Flip-Net is smaller than that of MLP, and is on par with LipsNet. (b) FlipNet has better action continuity than MLP and LipsNet. (c) FlipNet's action frequency is more distributed in the low-frequency range.

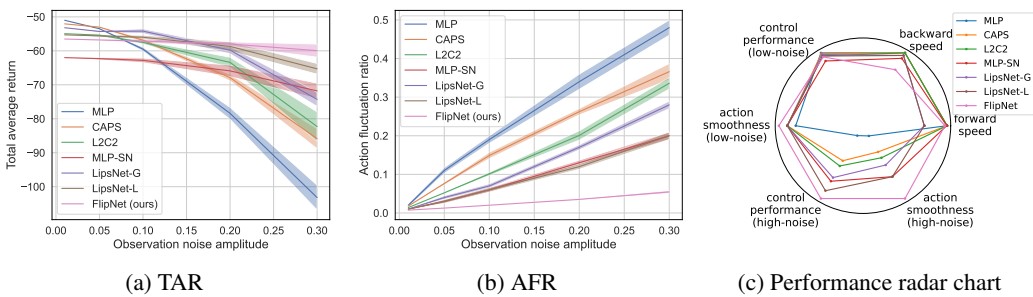

(a) TAR     (b) AFR     (c) Performance radar chart

Figure 8: **Performance comparison in double integrator environment.** (a) The TAR of FlipNet declines at the slowest rate when noise increases. (b) The AFR of FlipNet grows at the slowest rate when noise increases. (c) FlipNet has the best overall performance compared to previous works.

To further evaluate FlipNet, we set different observation noise amplitudes and compare with previous works. As Figure 8(a) shows, when noise increases, FlipNet maintains the highest TAR and its TAR declines at the slowest rate. As Figure 8(b) shows, when noise increases, FlipNet maintains the lowest AFR and its AFR grows at the slowest rate. We then compare the performance in high-noise environment, i.e. noise amplitude is 0.3. Compared to LipsNet-L, the previous SOTA network, FlipNet achieves an 8.2% increase in TAR and a 75.0% reduction in AFR. Therefore, FlipNet achieves a new SOTA performance with a significant advantage in action smoothness.

Furthermore, an ablation study for the two techniques in FlipNet is implemented in Appendix E, the sensitivity analysis for hyperparameters $\lambda_k$ and $\lambda_k$ is provided in Appendix F, and the sensitivity analysis for hyperparameter $N$ is provided in Appendix G. Additionally, policy networks' computational efficiency are evaluated in Appendix H, including the time usages of forward and backward propagations. Based on all the above results, a performance radar chart is depicted in Figure 8(c), which implies the overall performance of FlipNet is much better than previous works.

## 4.2 DEEPMIND CONTROL SUITE

The DeepMind Control Suite (DMControl) (Tassa et al., 2018) consist of several well-designed continuous control tasks. Currently, it stands as one of the most recognized benchmarks in the fields of RL and continuous control (Mu et al., 2022). In this paper, we focus on four of its environments: Cartpole, Reacher, Cheetah, and Walker. The visualization of these environments are shown in Figure 9, and more information are described in Appendix I.

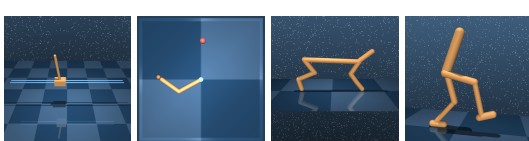

(a) Cartpole   (b) Reacher   (c) Cheetah   (d) Walker

Figure 9: **DeepMind control suite benchmark.**

We employ the Twin Delayed Deep Deterministic Policy Gradient (TD3) (Fujimoto et al., 2018), a model-free RL algorithm, for training. The hyperparameters for TD3 remain consistent across all environments, except for the coefficients $\lambda_k$, $\lambda_h$, and the length of historical observations $N$. All hyperparameters are listed in Appendix J. To evaluate comprehensively, networks are tested on both noise-free and noisy environments. Figure 10 visualizes the results in noisy environments. The learned filter matrix $H$ is visualized in Figure 24 to show the noise filtering ability of FlipNet. All results are summarized in Table 11 and 12, from which we can find that FlipNet has the highest TAR and the lowest AFR in all cases. For example, FlipNet increases the TAR by 3.4% and reduces the AFR by 35.5% in Walker environment compared to LipsNet, which is the previous SOTA network. Appendix K shows a comparison in Cartpole environment between FlipNet and reward penalty method. All these results imply that FlipNet has perfect action smoothness and noise robustness.

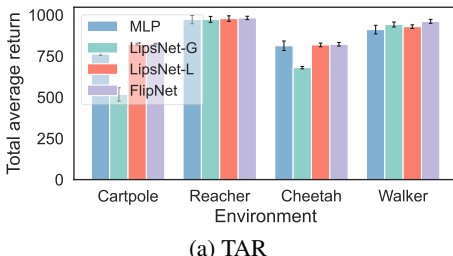
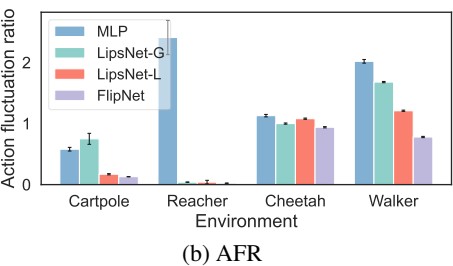

(a) TAR                     (b) AFR

Figure 10: **Performance comparison in DMControl.** The figure shows networks' TAR and AFR in noisy environments. FlipNet has the highest TAR and the lowest AFR in all cases.

## 4.3 MINI-VEHICLE DRIVING

Vehicle trajectory tracking is an important task in autonomous driving (Guan et al., 2022; Mu et al., 2020). To validate FlipNet in the real world, we conduct an experiment on physical vehicles. As Figure 26 shows, the vehicle moves by two differential wheels, aiming to track reference trajectory and velocity while avoiding obstacle. The observations and actions are listed in Table 15. We set up four diverse scenarios, as described in Table 1 and visualized in Figure 28. Detailed introduction

of the vehicle, control mode, and scenarios are described in Appendix L. The Distributional Soft Actor-critic (DSAC) (Duan et al., 2021), a model-free RL algorithm, is used for training. The tests in all scenarios are accomplished by the same networks. For real-world highway vehicles, RL observations rely on perception results where sensor noise is amplified by perception algorithms. To precisely simulate such scenario, we assigned various noise amplitudes. A test video is available [4].

Table 1: **Scenario descriptions in mini-vehicle driving environment.**

| Scenario | RL robot | Obstacle robot | Description |
|---|---|---|---|
| 1 | go straight | static | RL robot goes straight and avoids static obstacle |
| 2 | go straight | moving | RL robot goes straight and avoids moving obstacle |
| 3 | turn left | moving | RL robot turns left and avoids moving obstacle |
| 4 | go straight | aggressive | RL robot goes straight and avoids aggressive obstacle |

In scenario 3 with 10 times noise, the results are shown in Figure 11, its video snapshots are recorded in Figure 12. The RL robot successfully tracks the reference trajectory and avoids obstacle by slightly shifting to yield. As shown in Figure 11(d)(e), it is evident that FlipNet produces much smoother control actions than MLP. The smoother actions result in smoother vehicle states, i.e. speed and yaw rate, which are shown in Figure 11(b)(c). These results consistently hold true across all scenarios, as illustrated in Appendx M. Furthermore, in scenario 4 with 10 times noise, the RL robot driven by MLP crashes while FlipNet successfully completes the task, as shown in Figure 47 and 48.

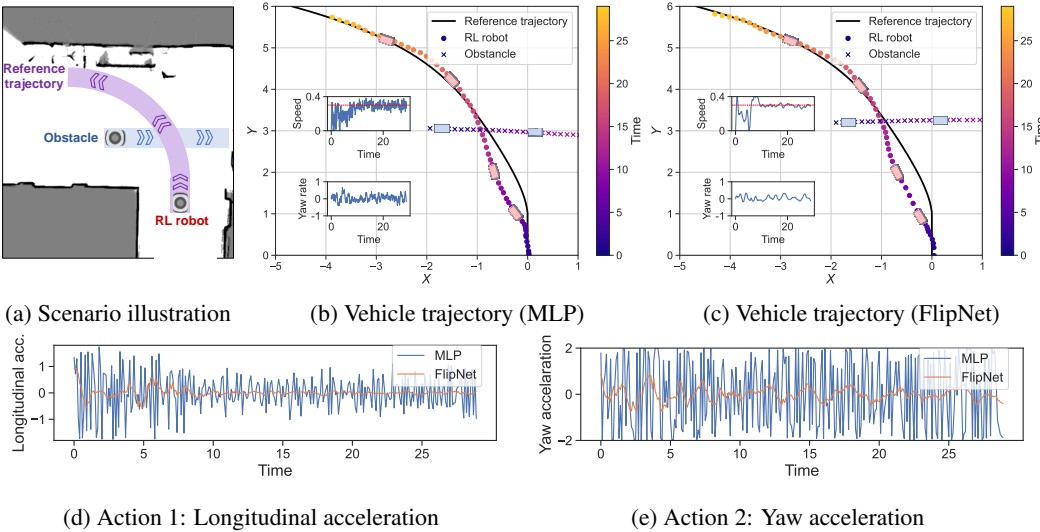

(a) Scenario illustration     (b) Vehicle trajectory (MLP)     (c) Vehicle trajectory (FlipNet)

(d) Action 1: Longitudinal acceleration     (e) Action 2: Yaw acceleration

Figure 11: **Result of scenario 3.** The noise amplitude is 10. (a) The RL robot aims to turn left. (b,c) The vehicle states and trajectories produced by MLP and FlipNet. (d,e) The control actions produced by MLP and FlipNet. FlipNet produces much smoother control actions than MLP.

The learned filter matrix $H$ is visualized in Figure 13 to show the noise filtering ability of FlipNet. Figure 13(a) and (b) show the frequency distributions of observation in noise-free and noisy environments. Figure 13(c) implies that the learned filter matrix mainly focus on the frequencies containing observation information, and rarely focus on the frequencies containing noises. In other words, FlipNet can automatically extract the important frequencies and filter out the noise frequencies.

The average TAR and AFR for the first three scenarios are depicted in Figure 14. As Figure 14(a) shows, when noise increases, FlipNet maintains the highest TAR and its TAR declines much slower than MLP's. As Figure 14(b) shows, when noise increases, FlipNet maintains the lowest AFR and

---

[4]Project page: https://iclr-anonymous-2025.github.io/FlipNet

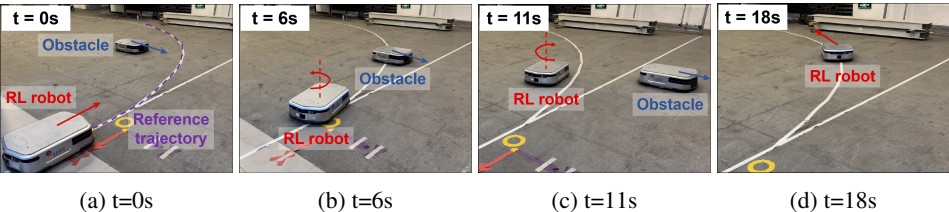

(a) t=0s      (b) t=6s      (c) t=11s      (d) t=18s

Figure 12: **Snapshots of scenario 3.** The figures are the video snapshots of Figure 11(c). The RL robot first shifts to the left to yield, then resumes tracking the reference trajectory.

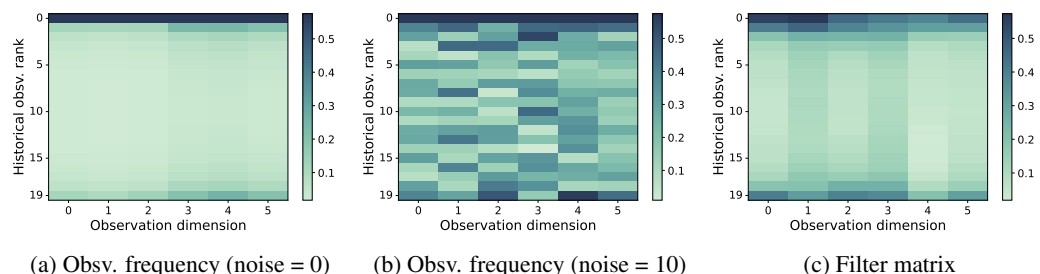

(a) Obsv. frequency (noise = 0)  (b) Obsv. frequency (noise = 10)  (c) Filter matrix

Figure 13: **Filter matrix and observation frequency in mini-vehicle driving environment.** The color in (a) and (b) represents the intensity of frequency. The color in (c) represents the magnitude of elements in matrix $H$. The color distribution in (c) implies FlipNet can automatically extract the important frequencies and filter out the noise frequencies.

its AFR grows much slower than MLP's. In the high-noise environment, i.e. noise amplitude is 20, FlipNet achieves an 5.9% increase in TAR and a 90.0% reduction in AFR. The results imply FlipNet has much better action smoothness and noise robustness. More results can be found in Appendix M.

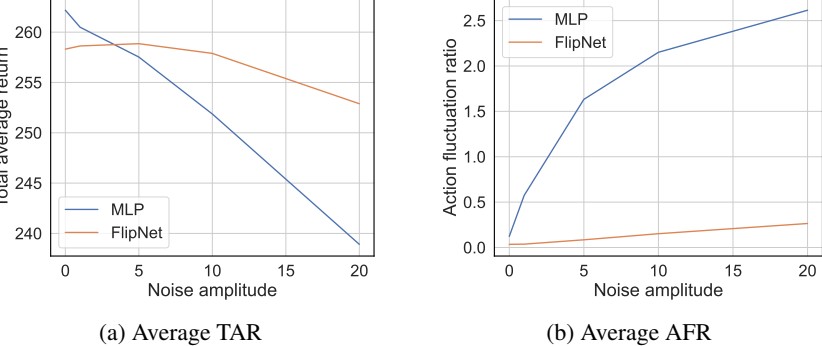

(a) Average TAR          (b) Average AFR

Figure 14: **Performance trend in mini-vehicle driving environment.** The curves show the average TAR and AFR for the first three scenarios. (a) The TAR of FlipNet declines much slower than MLP's when noise increases. (b) The AFR of FlipNet grows much slower than MLP's when noise increases. It implies that FlipeNet has much better action smoothness and noise robustness.

## 5 CONCLUSION

In this paper, we identify the two fundamental reasons causing action fluctuation, and propose the Fourier Lipschitz Smooth Policy Network (FlipNet). FlipNet adopts two innovative techniques to directly tackle the two reasons. Firstly, we prove the Jacobian norm is an approximation of Lipschitz constant and introduce the Jacobian regularization to enhance the policy smoothness. Secondly, we introduce a Fourier filter layer to deal with observation noise. The layer includes a trainable filter matrix that can automatically extract valuable frequencies and suppress noise frequencies. FlipNet can be easily used as actor networks in most RL algorithms. Simulated and real-world experiments show that FlipeNet has excellent action smoothness and noise robustness, achieving a new SOTA performance. We hope that the research could contribute to the applications of RL in the real world.

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

## A    Theoretical Results

**Lemma A.1** (Equivalent Form of Lipschitz Constant). *Suppose $f : \mathbb{R}^n \to \mathbb{R}^m$ is a continuously differential neural network. Then its local Lipschitz constant $K(x)$ has an equivalent form besides equation (5):*

$$K(x) = \max_{x' \in \mathcal{B}(x,\rho)} \|\nabla f(x')\|. \tag{11}$$

*Proof.* We assume the real local Lipschitz constant of $f$ over $\mathcal{B}(x,\rho)$ is $K_x$, which means $K_x = \max_{x_1,x_2 \in \mathcal{B}(x,\rho)} \frac{\|f(x_1)-f(x_2)\|}{\|x_1-x_2\|}$. The following proof is similar to that of (Song et al., 2023).

**(a)** Firstly, we prove that $\|\nabla f(x')\| \le K_x, \forall x' \in \mathcal{B}(x,\rho)$. Because the local Lipschitz constant is $K_x$, we know that

$$\|f(x_1) - f(x_2)\| \le K_x \|x_1 - x_2\|, \ \forall x_1, x_2 \in \mathcal{B}(x,\rho). \tag{12}$$

Let $h(t) = f(x' + t \cdot v)$ where $x' \in \mathcal{B}(x,\rho)$, $t \in \mathbb{R}$, and $v \in \mathbb{R}^n$, then its first-order derivative function is $h'(t) = \nabla f(x' + t \cdot v) \cdot v$. From the *Newton-Leibniz formula*, we know

$$h(\alpha) - h(0) = \int_0^\alpha h'(t) \, \mathrm{d}t,$$

which means

$$f(x' + \alpha \cdot v) - f(x') = \int_0^\alpha \nabla f(x' + t \cdot v) \cdot v \, \mathrm{d}t.$$

By taking the 2-norm on both sides and considering the condition (12), we get

$$\left\| \int_0^\alpha \nabla f(x' + t \cdot v) \, \mathrm{d}t \cdot v \right\| = \|f(x' + \alpha \cdot v) - f(x')\|$$

$$\le \alpha K_x \|v\|.$$

Divide $\alpha$ on both sides then let $\alpha \to 0^+$, get

$$\|\nabla f(x') \cdot v\| \le K_x \|v\|, \ \forall v.$$

From the definition of matrix norm, we know

$$\|\nabla f(x')\| = \max_{v \ne 0} \frac{\|\nabla f(x') \cdot v\|}{\|v\|} \le K_x, \forall x' \in \mathcal{B}(x,\rho).$$

**(b)** Secondly, we prove that $\max_{x' \in \mathcal{B}(x,\rho)} \|\nabla f(x')\| \ge K_x$. Let $h(t) = f(x_1 + t(x_2 - x_1))$ where $t \in (0,1)$ and $x_1, x_2 \in \mathcal{B}(x,\rho)$, then its first-order derivative function is $h'(t) = \nabla f(x_1 + t(x_2 - x_1)) \cdot (x_2 - x_1)$. From the *Newton-Leibniz formula*, we know

$$h(1) - h(0) = \int_0^1 h'(t) \, \mathrm{d}t,$$

which means

$$f(x_2) - f(x_1) = \int_0^1 \nabla f(x_1 + t(x_2 - x_1)) \cdot (x_2 - x_1) \, \mathrm{d}t$$

$$= \left( \int_0^1 \nabla f(x_1 + t(x_2 - x_1)) \, \mathrm{d}t \right) (x_2 - x_1).$$

Take the 2-norm on both sides, get

$$\|f(x_2) - f(x_1)\| = \left\| \left( \int_0^1 \nabla f(x_1 + t(x_2 - x_1)) \, \mathrm{d}t \right) (x_2 - x_1) \right\|$$

$$\le \left\| \int_0^1 \nabla f(x_1 + t(x_2 - x_1)) \, \mathrm{d}t \right\| \|x_2 - x_1\|$$

$$\le \left( \int_0^1 \|\nabla f(x_1 + t(x_2 - x_1))\| \, \mathrm{d}t \right) \|x_2 - x_1\|$$

$$\le \max_{x' \in \mathcal{B}(x,\rho)} \|\nabla f(x')\| \cdot \|x_2 - x_1\|.$$

Therefore,

$$\max_{x' \in \mathcal{B}(x,\rho)} \|\nabla f(x')\| \geq \frac{\|f(x_1) - f(x_2)\|}{\|x_1 - x_2\|}, \ \forall x_1, x_2 \in \mathcal{B}(x,\rho),$$

which means

$$\max_{x' \in \mathcal{B}(x,\rho)} \|\nabla f(x')\| \geq \max_{x_1,x_2 \in \mathcal{B}(x,\rho)} \frac{\|f(x_1) - f(x_2)\|}{\|x_1 - x_2\|}$$
$$= K_x.$$

Considering both (a) and (b), we know that $\|\nabla f(x')\| \leq K_x, \forall x' \in \mathcal{B}(x,\rho)$ and $\max_{x' \in \mathcal{B}(x,\rho)} \|\nabla f(x')\| \geq K_x$. Therefore, we can conclude that $\max_{x' \in \mathcal{B}(x,\rho)} \|\nabla f(x')\| = K_x$. $\qquad\square$

**Theorem A.2** (Lipschitz's Jacobian Approximation). *For a continuously differential neural network $f : \mathbb{R}^n \to \mathbb{R}^m$, the Jacobian norm $\|\nabla_x f\|$ is an approximation of the local Lipschitz constant of $f$ on the infinitesimal neighborhood of $x$, i.e. $K(x) \approx \|\nabla_x f\|$.*

*Proof.* By Definition 3.1 and Lemma A.1, we know that

$$K(x) = \max_{x_1,x_2 \in \mathcal{B}(x,\rho)} \frac{\|f(x_1) - f(x_2)\|}{\|x_1 - x_2\|}$$
$$= \max_{x' \in \mathcal{B}(x,\rho)} \|\nabla f(x')\|$$
$$= \max_{\delta \in \mathcal{B}(0,\rho)} \|\nabla f(x + \delta)\|.$$

By conducting the first-order Taylor expansion for $\|\nabla f(x + \delta)\|$, we get that

$$K(x) = \max_{\delta \in \mathcal{B}(0,\rho)} \left[ \|\nabla_x f(x)\| + (\nabla_x \|\nabla_x f(x)\|)^\top \delta + o(\delta) \right]$$
$$= \|\nabla_x f(x)\| + \max_{\delta \in \mathcal{B}(0,\rho)} \left[ (\nabla_x \|\nabla_x f(x)\|)^\top \delta + o(\delta) \right].$$

We know that $\rho \to 0$ because $\mathcal{B}(0,\rho)$ is a infinitesimal neighborhood of $x$, then $\delta \to 0$ holds. Therefore, $K(x) \approx \|\nabla_x f(x)\|$. $\qquad\square$

## B  PSEUDOCODE OF FLIPNET

---
**Algorithm 1** Forward and backward propagation of FlipNet
---
**Require:** historical observations $o_t, o_{t-1}, \cdots, o_{t-N+1}$, actor loss $\mathcal{L}$, network parameter $\theta$.

1: /* Forward propagation */
2: $x \leftarrow [o_t \ o_{t-1} \ \cdots \ o_{t-N+1}]^\top$
3: $X \leftarrow \mathrm{FFT}(x)$
4: $\tilde{X} = \mathrm{symmetrize}(X_{\mathrm{half}} \odot H)$.
5: $\tilde{x} \leftarrow \mathrm{IFFT}\left(\tilde{X}\right)$
6: $\tilde{o}_t \leftarrow$ the first row in $\tilde{x}$
7: $a_t \leftarrow f(\tilde{o}_t)$

8: /* Backward propagation */
9: $\mathcal{L}'' \leftarrow \mathcal{L} + \lambda_k \|\nabla_{\tilde{o}_t} f\| + \lambda_h \|H\|_F$
10: $\theta_{\mathrm{new}} \leftarrow \theta - \eta \nabla_\theta \mathcal{L}''$

**Ensure:** control action $a_t$, updated network parameter $\theta_{\mathrm{new}}$.

---

## C  FUNDAMENTAL REASONS OF ACTION FLUCTUATION

**Non-smoothness of policy network.** A non-smooth policy network means that RL fits a non-smooth policy function mapping from the state to control action. The mapping function has significant output differences even the inputs are closely adjacent. Consequently, when the state changes with time, a non-smooth action sequence is generated. Figure 15 visualizes the effect of policy non-smoothness.

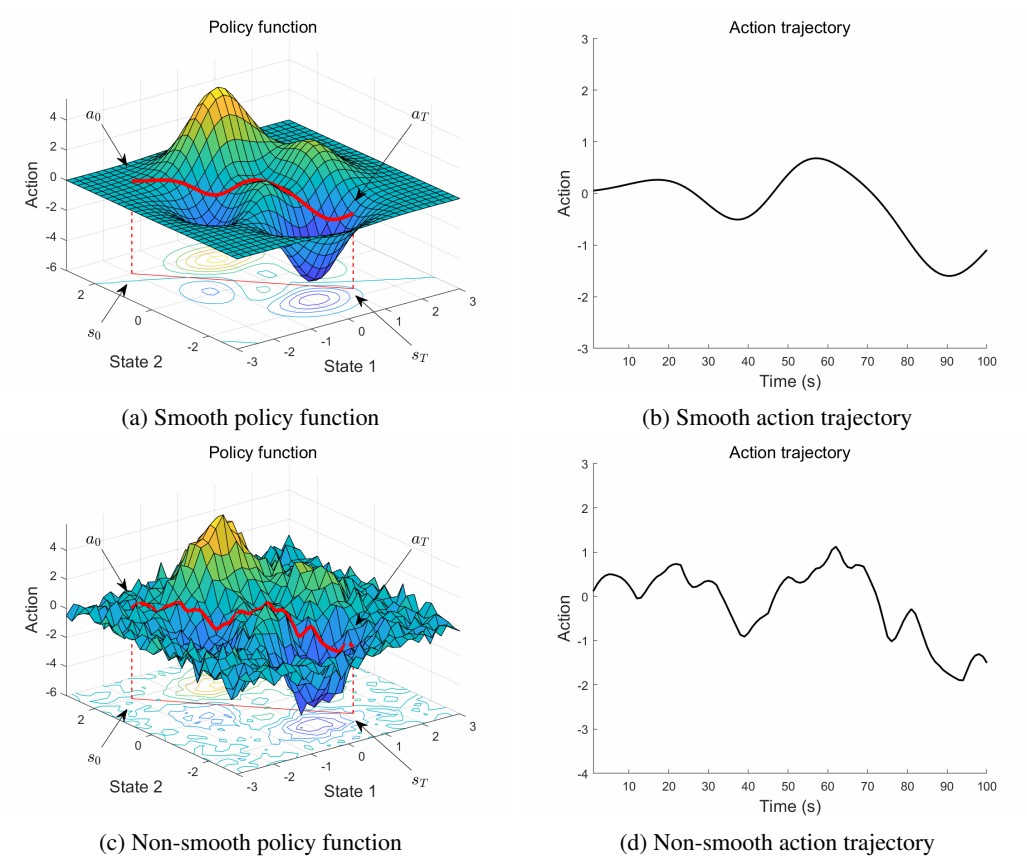

(a) Smooth policy function

(b) Smooth action trajectory

(c) Non-smooth policy function

(d) Non-smooth action trajectory

Figure 15: **Effect of policy non-smoothness.**

**Existence of observation noise.** The noise results in the discontinuous changes in observations, making the actions produced by the policy network at the adjacent time stamps erratically differ. Even if the policy function fitted by the policy network is smooth enough, actions can still be fluctuated because of the erratic observation noise.

## D  DOUBLE INTEGRATOR: DETAILED IMPLEMENTATION AND RESULTS

The double integrator is a classic control task with linear dynamics and quadratic cost function, namely linear quadratic (LQ) control task. The environment used in this paper is a particle-moving environment. We train in noise-free environment and test in noisy environment with various noise level to comprehensively evaluate policy networks. We use a model-based RL algorithm, INFADP (Li, 2023), to train different policy networks including MLP (Rumelhart et al., 1986), MLP-SN (Takase et al., 2020), LipsNet-G (Song et al., 2023), LipsNet-L (Song et al., 2023), and FlipNet. The hyperparameters for INFADP are listed in Table 2.

We set 5 different observation noise amplitudes and compare the performances of MLP, MLP-SN, LipsNet-G, LipsNet-L, and FlipNet. Table 3 and Table 4 summarize the TAR and AFR, respectively. Figure 8 shows the variation trends of TAR and AFR as the noise increases. As shown in Figure 8(a),

the TAR of FlipNet decreases much slower than that of the other networks. As shown in Figure 8(b), the AFR of FlipNet increases much slower than that of the other networks. These results indicate that FlipNet has superior action smoothness and noise robustness compared to previous works.

Table 2: **Hyperparameters for INFADP.**

| Hyperparameter | Value |
|---|---|
| Replay buffer capacity | 100000 |
| Buffer warm-up size | 1000 |
| Batch size | 64 |
| Discount $\gamma$ | 0.99 |
| Target network soft-update rate $\tau$ | 0.2 |
| Initial random interaction steps | 0 |
| Interaction steps per iteration | 8 |
| Network update times per iteration | 1 |
| Prediction step | 1 |
| Action bound | $[-5, 5]$ |
| Exploration noise std. deviation | 0 |
| Hidden layers in subnetwork $f$ | $[64, 64]$ |
| Activations in subnetwork $f$ | ReLU |
| Hidden layers in critic network | $[64, 64]$ |
| Activations in critic network | ReLU |
| Optimizer | Adam |
| Actor learning rate | $3 \cdot 10^{-4}$ |
| Critic learning rate | $8 \cdot 10^{-4}$ |
| length of historical obsv. $N$ | 8 |
| coefficient $\lambda_k$ | 0.01 |
| coefficient $\lambda_h$ | 1 |

Table 3: **Comparison of TAR on double integrator environment.** The observation noise in each dimension is distributed in $U(-\sigma, \sigma)$. The data in this table is visualized in Figure 8(a).

| Noise | Methods | | | | | | |
|---|---|---|---|---|---|---|---|
| $\sigma$ | MLP | CAPS | L2C2 | MLP-SN | LipsNet-G | LipsNet-L | FlipNet |
| 0.01 | **-51.0** $_{\pm 0.1}$ | -52.1 $_{\pm 0.1}$ | -55.0 $_{\pm 0.1}$ | -62.0 $_{\pm 0.1}$ | -53.2 $_{\pm 0.1}$ | -55.3 $_{\pm 0.1}$ | -56.5 $_{\pm 0.1}$ |
| 0.05 | -53.5 $_{\pm 0.2}$ | **-53.0** $_{\pm 0.2}$ | -55.4 $_{\pm 0.4}$ | -62.3 $_{\pm 0.4}$ | -54.3 $_{\pm 0.3}$ | -55.6 $_{\pm 0.4}$ | -56.8 $_{\pm 0.2}$ |
| 0.1 | -59.5 $_{\pm 0.6}$ | -56.9 $_{\pm 0.6}$ | -57.5 $_{\pm 0.5}$ | -62.8 $_{\pm 0.7}$ | **-54.2** $_{\pm 0.7}$ | -56.0 $_{\pm 0.6}$ | -57.1 $_{\pm 0.6}$ |
| 0.2 | -78.4 $_{\pm 1.8}$ | -67.9 $_{\pm 1.0}$ | -63.4 $_{\pm 1.5}$ | -65.9 $_{\pm 1.7}$ | -59.8 $_{\pm 1.1}$ | -58.7 $_{\pm 1.4}$ | **-57.9** $_{\pm 0.6}$ |
| 0.3 | -103.2 $_{\pm 3.7}$ | -85.9 $_{\pm 3.0}$ | -82.3 $_{\pm 4.4}$ | -71.8 $_{\pm 2.3}$ | -74.3 $_{\pm 2.1}$ | -65.3 $_{\pm 1.6}$ | **-59.9** $_{\pm 1.9}$ |

Table 4: **Comparison of AFR on double integrator environment.** The observation noise in each dimension is distributed in $U(-\sigma, \sigma)$. The data in this table is visualized in Figure 8(b).

| Noise | Methods | | | | | | |
|---|---|---|---|---|---|---|---|
| $\sigma$ | MLP | CAPS | L2C2 | MLP-SN | LipsNet-G | LipsNet-L | FlipNet |
| 0.01 | 0.02 $_{\pm 0.01}$ | 0.01 $_{\pm 0.01}$ | 0.01 $_{\pm 0.01}$ | 0.01 $_{\pm 0.01}$ | 0.01 $_{\pm 0.01}$ | 0.01 $_{\pm 0.01}$ | **0.00** $_{\pm 0.01}$ |
| 0.05 | 0.11 $_{\pm 0.01}$ | 0.07 $_{\pm 0.01}$ | 0.05 $_{\pm 0.01}$ | 0.03 $_{\pm 0.01}$ | 0.04 $_{\pm 0.01}$ | 0.03 $_{\pm 0.01}$ | **0.01** $_{\pm 0.01}$ |
| 0.1 | 0.19 $_{\pm 0.01}$ | 0.15 $_{\pm 0.01}$ | 0.10 $_{\pm 0.01}$ | 0.06 $_{\pm 0.01}$ | 0.07 $_{\pm 0.01}$ | 0.06 $_{\pm 0.01}$ | **0.02** $_{\pm 0.01}$ |
| 0.2 | 0.34 $_{\pm 0.02}$ | 0.26 $_{\pm 0.01}$ | 0.20 $_{\pm 0.01}$ | 0.13 $_{\pm 0.01}$ | 0.17 $_{\pm 0.01}$ | 0.12 $_{\pm 0.01}$ | **0.03** $_{\pm 0.01}$ |
| 0.3 | 0.48 $_{\pm 0.02}$ | 0.37 $_{\pm 0.02}$ | 0.33 $_{\pm 0.02}$ | 0.20 $_{\pm 0.01}$ | 0.28 $_{\pm 0.01}$ | 0.20 $_{\pm 0.01}$ | **0.05** $_{\pm 0.01}$ |

# E    ABLATION STUDY FOR TWO TECHNIQUES

In this appendix, we implement ablation study for the two techniques proposed in our paper, i.e. Jacobian regularization and Fourier filter layer. The two techniques respectively tackle the two fundamental reasons of action fluctuation, as described in Section 3.1. The Jacobian regularization enhances the smoothness of policy network by introducing the Jacobian norm in actor loss function. Similarly, the Fourier filter layer enhance the noise robustness of policy network by introducing the Frobenius norm of the filter matrix in actor loss function. The resulted actor loss is illustrated in Equation 9:

$$\mathcal{L}'' = \mathcal{L} + \lambda_k \left\| \nabla f \right\| + \lambda_h \left\| H \right\|_F .$$

In order to validate the effectiveness of each technique, the two coefficients $\lambda_k$ and $\lambda_h$ are set to zero in turn. The performance result on double integrator environment is shown in Figure 16. The performance result on DMControl's Cheetah and Walker environments is shown in Table 5. These results shows that setting either coefficient to zero will lead to a rapid decrease in TAR and a rapid increase in AFR when the noise increases. It indicates that both the Jacobian regularization and Fourier filter layer are effective techniques and they are both indispensable.

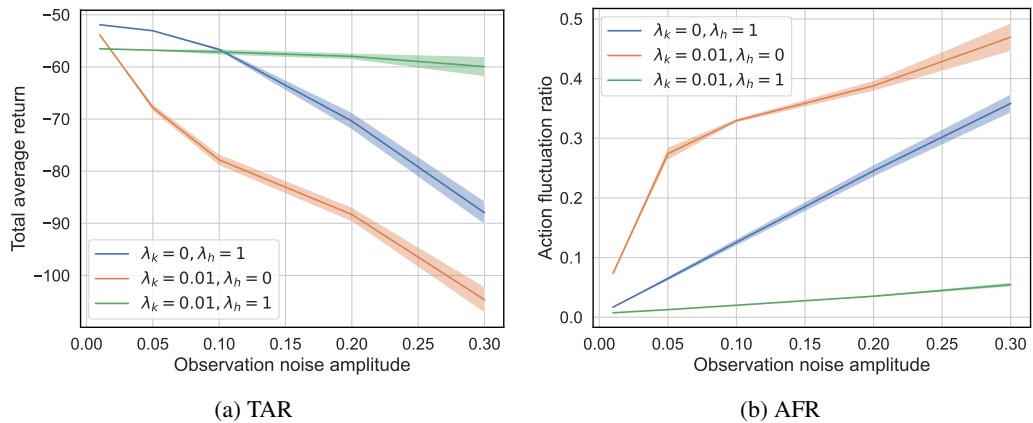

(a) TAR                                                    (b) AFR

Figure 16: **Ablation study for Jacobian regularization and Fourier filter layer on double integrator environment.**

Table 5: **Ablation study on Cheetah and Walker.** The result shows that setting either coefficient to zero will lead to an increase in AFR, which indicates the two techniques are all effective and indispensable.

| Environment | $\lambda_k$ | $\lambda_h$ | Total average return | Action fluctuation ratio |
|---|---|---|---|---|
| | $10^{-3}$ | $10^{-3}$ | **822** $_{\pm 11}$ | **0.94** $_{\pm 0.01}$ |
| Cheetah | 0 | $10^{-3}$ | 822 $_{\pm 15}$ | 1.08 $_{\pm 0.02}$ |
| | $10^{-3}$ | 0 | 821 $_{\pm 17}$ | 1.21 $_{\pm 0.02}$ |
| | $10^{-2}$ | $10^{-3}$ | **961** $_{\pm 12}$ | **0.78** $_{\pm 0.01}$ |
| Walker | 0 | $10^{-3}$ | 958 $_{\pm 15}$ | 0.98 $_{\pm 0.02}$ |
| | $10^{-2}$ | 0 | 940 $_{\pm 14}$ | 1.89 $_{\pm 0.01}$ |

# F  SENSITIVITY ANALYSIS FOR $\lambda_k$ AND $\lambda_h$

In this appendix, we provide the sensitivity analysis for the hyperparameters $\lambda_k$ and $\lambda_h$. We design experiments to demonstrate that the two hyperparameters have low sensitivity, making FlipNet convenient for tuning and easy to use.

Similar to the approach in Appendix E, we fix one hyperparameter and then vary the other to observe the changes in TAR and AFR on double integrator environment. As shown in Figure 17, when $\lambda_h$ is fixed at 1 and $\lambda_k$ varies between 0.001, 0.01, and 0.1, the performance differences are significant. However, when $\lambda_h$ is fixed at 1 and $\lambda_k$ varies between 0.01 and 0.02, the performances are essentially consistent. A similar phenomenon can also be found for hyperparameter $\lambda_h$, as shown in Figure 18. When $\lambda_k$ is fixed at 0.01 and $\lambda_h$ varies between 0.1, 1 and 10, the performance differences are significant. However, when $\lambda_h$ is fixed at 1 and $\lambda_k$ varies between 1 and 2, the performances are essentially consistent.

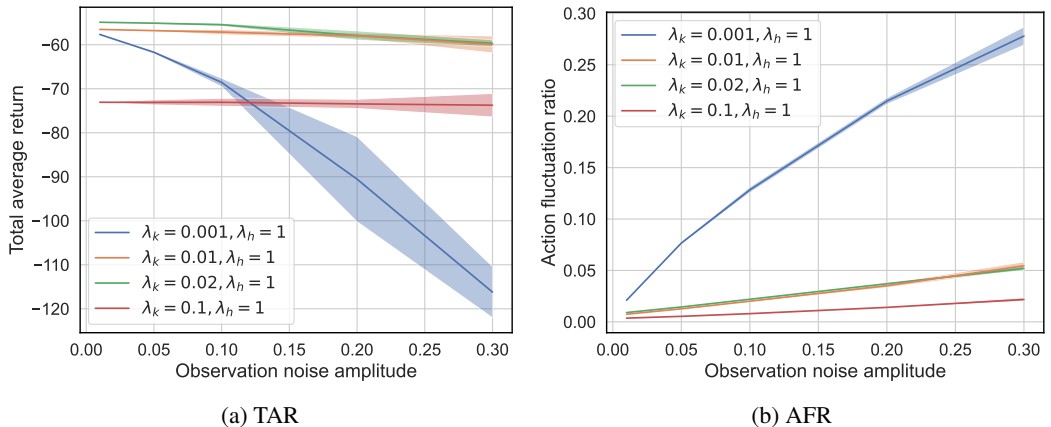

(a) TAR                    (b) AFR

Figure 17: **Sensitivity analysis for $\lambda_k$.**

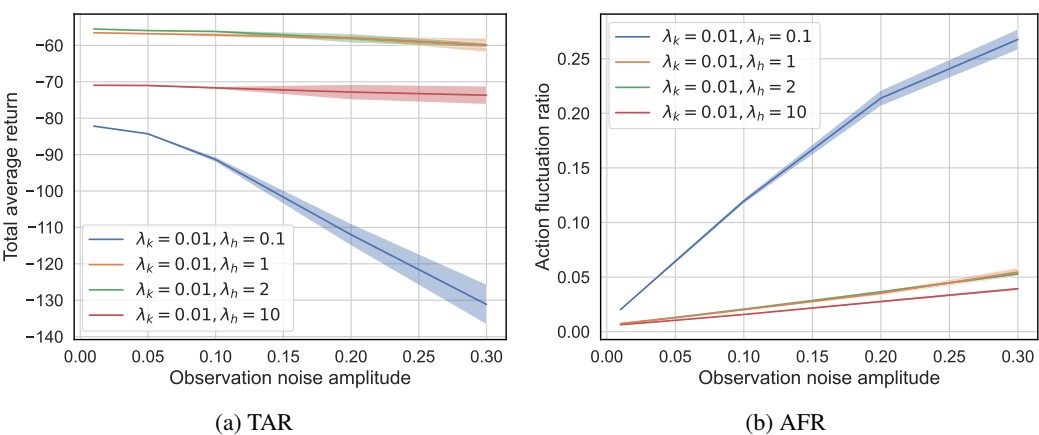

(a) TAR                    (b) AFR

Figure 18: **Sensitivity analysis for $\lambda_h$.**

The above results imply that the hyperparameter $\lambda_k$ and $\lambda_h$ have low sensitivity. Only the magnitude of hyperparameters have a significant impact on performance, while changing the hyperparameters within an appropriate magnitude has a minimal effect on performance. Therefore, when tuning parameters, the user only need to set an appropriate magnitude. It makes FlipNet convenient for tuning and easy to use.

# G    SENSITIVITY ANALYSIS FOR $N$

In this appendix, we provide the sensitivity analysis for hyperparameter $N$, which represents the length of historical observations. We design experiments to demonstrate that $N$ exhibits low sensitivity when the length of historical observations is sufficiently long, making FlipNet convenient for tuning and easy to use.

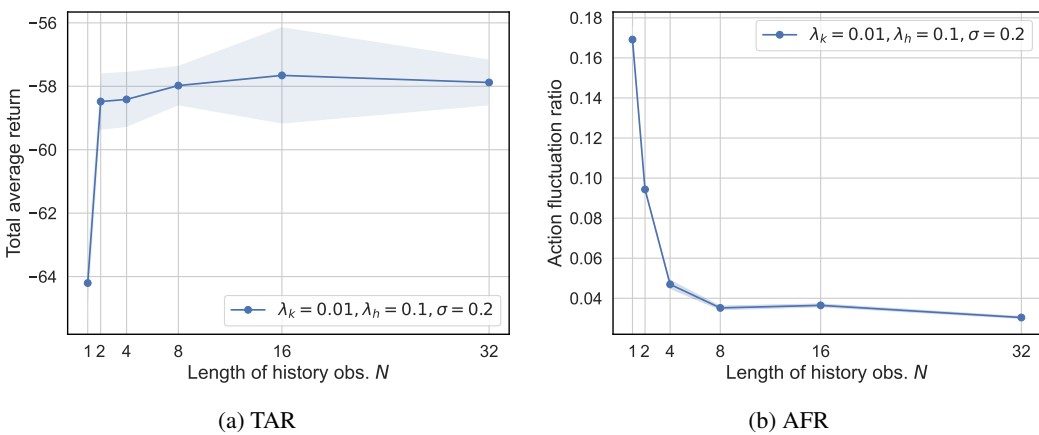

(a) TAR                    (b) AFR

Figure 19: **Sensitivity analysis for $N$.**

The values of $N$ are set to range from 1 to 32 in the double integrator environment. The Figure 19(a) and (b) show the trend of TAR and AFR when $N$ changes. The result shows that the performance no longer improves once $N$ exceeds a threshold, suggesting a low sensitivity. Therefore, users only need to set a relatively large value of $N$, making FlipNet very convenient for tuning.

# H    COMPUTATIONAL EFFICIENCY ANALYSIS

To evaluate the computational efficiency, we provide the detailed forward and backward processing time of policy networks. All policy networks are from the network trained in double integrator environment. This analysis is implemented on AMD Ryzen Threadripper 3960X 24-Core Processor. For MLP-SN, the number of power iterations is set to 1, whose time usage is included in the backward stage. Similarly, the computation times for Jacobian norm and Frobenius norm in FlipeNet are included in the backward stage. The length of historical observations used in FlipeNet is 8.

The results are summarized in Table 6. Compared to the previous SOTA network LipsNet, FlipNet has significantly faster speed for forward propagation. This allows FlipNet to be applied in high real-time tasks. We acknowledge that backward propagation speed of FlipNet is relatively slow, but we have devised a solution to accelerate this in future work by using multiple forward propagation and zero-order gradient estimation to compute the Jacobian matrix.

Table 6: Forward and backward **propagation time comparison**.

| Settings | | Policy network | | | |
|---|---|---|---|---|---|
| Propagation | Batch size | MLP | MLP-SN | LipsNet-L | FlipNet |
| forward | 1 | 0.10 ms | 0.11 ms | 0.75 ms | 0.16 ms |
| | 100 | 0.11 ms | 0.12 ms | 1.41 ms | 0.25 ms |
| backward | 1 | 0.17 ms | 0.76 ms | 0.45 ms | 1.98 ms |
| | 100 | 0.28 ms | 0.89 ms | 0.73 ms | 2.48 ms |

Since network propagation times constitute only a part of the overall RL training process, we next compare the total training wall-clock times. Table 7 presents the wall-clock times for 1M iterations of TD3 on the DMControl environments. The results show that, on average, the training time of

FlipNet is 1.6 times that of MLP. The difference in wall-clock time is not as significant as the difference in backward time shown in Table 6, as RL algorithms involve additional time-consuming steps beyond backward, such as sampling and evaluation.

Table 7: **Training wall-clock time comparison.** The data show the wall-clock times used for 1M iterations in TD3. On average, the training time of FlipNet is 1.6 times that of MLP.

| Network | Env | | | | Total |
|---------|---------|---------|---------|--------|---------|
| | Cartpole | Reacher | Cheetah | Walker | |
| MLP | 120 min | 118 min | 128 min | 125 min | 491 min |
| FlipNet | 194 min | 195 min | 206 min | 204 min | 799 min |

In conclusion, FlipNet's forward time is under 0.2 ms, making it suitable for real-time applications. While the training wall-clock time shows a slight increase, it remains acceptable and has clear pathways for future optimization.

## I DEEPMIND CONTROL SUITE BENCHMARK

The DeepMind Control Suite (DMControl) (Tassa et al., 2018) encompasses a collection of meticulously crafted continuous control tasks. These environments feature consistent structures, rewards that are both interpretable and normalized, facilitating a more straightforward comparison of performance across different algorithms. Developed in Python and leveraging the MuJoCo physics engine (Todorov et al., 2012), DMControl currently stands as one of the most esteemed benchmarks for evaluating RL and continuous control tasks.

In DMControl, the term "domain" denotes a specific physical model, whereas a "task" corresponds to an instantiation of that model with a defined Markov Decision Process (MDP) structure. For instance, within the cartpole domain, the distinction between the `swingup` and `balance` tasks lies in the initial orientation of the pole: it is initialized pointing downward in the `swingup` task and upward in the `balance` task, respectively. In the following figures, we provide detailed descriptions of the domains used in this paper, with each domain's name followed by a tuple of three integers that denote the dimensions of the state, action, and observation spaces, respectively, formatted as $\left(\dim\left(\mathcal{S}\right), \dim\left(\mathcal{A}\right), \dim\left(\mathcal{O}\right)\right)$.

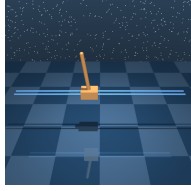

Figure 20: **Cartpole(4, 1, 5):** This domain features a cart connected to a pole via an unactuated joint. It encompasses a set of four distinct tasks. In the context of our experimental setup, we focus on the `swingup` task. Here, the pole is initially positioned downward, and the objective is to apply appropriate forces to the cart to swing the pole upward and maintain its upright position.

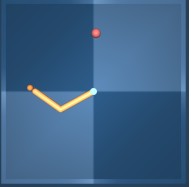

Figure 21: **Reacher(4, 2, 7):** This domain comprises two interconnected poles with a sphere whose initial position is randomly determined. One end of the linked poles is anchored at the origin of the coordinate space, while the other remains free to move. The domain offers two distinct tasks, and we focus on the `easy` task. The task requires the application of forces to the pendulum to ensure that its endpoint remains within the red area at all times.

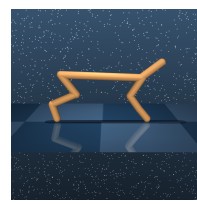

Figure 22: **Cheetah**(18, 6, 17): This domain features a planar bipedal and it is able to crawl forward by its two legs. It involves a single task, namely the `run` task. In the initial state of the environment, the agent's pose is random, typically in a non-standing position. In this task, the challenge is to control the planar biped to achieve an upright standing position and subsequently propel it forward into a running motion with a targeted forward velocity.

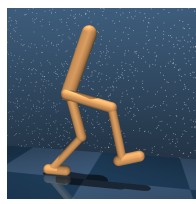

Figure 23: **Walker**(18, 6, 24): This domain includes a planar walker. This environment simulates a simple locomotion task of humans, with the agent possessing two legs and advancing in an upright posture. It comprises three distinct tasks, and our experiment focus the `walk` task. In this task, the objective is to control the walker to maintain an upright torso posture, achieve the specified torso height, and maintain a consistent forward velocity.

## J DMCONTROL: DETAILED IMPLEMENTATION AND RESULTS

We employ the Twin Delayed Deep Deterministic Policy Gradient (TD3) (Fujimoto et al., 2018), a model-free RL algorithm, to train on DMControl. The hyperparameters for TD3 remain consistent across all environments, except for the coefficients $\lambda_k$, $\lambda_h$, and the length of historical observations $N$. The hyperparameters for TD3 are listed in Table 8. The environment-related hyperparameters are listed in Table 9.

Table 8: **Hyperparameters for TD3.**

| Hyperparameter | Value |
|---|---|
| Replay buffer capacity | 1000000 |
| Buffer warm-up size | 1000 |
| Batch size | 100 |
| Discount $\gamma$ | 0.99 |
| Target network soft-update rate $\tau$ | 0.005 |
| Target noise | 0.2 |
| Target noise limit | 0.5 |
| Exploration noise std. deviation | 0.1 |
| Policy delay times | 2 |
| Initial random interaction steps | 25000 |
| Interaction steps per iteration | 50 |
| Network update times per iteration | 50 |
| Hidden layers in subnetwork $f$ | $[64, 64]$ |
| Activations in subnetwork $f$ | ReLU |
| Hidden layers in critic network | $[64, 64]$ |
| Activations in critic network | ReLU |
| Optimizer | Adam |
| Actor learning rate | $1 \cdot 10^{-3}$ |
| Critic learning rate | $1 \cdot 10^{-3}$ |

To evaluate comprehensively, networks are tested on both noise-free and noisy environments. For noisy environments, the noise amplitudes are listed in Table 10. We compare FlipNet with MLP, LipsNet-G, LipsNet-L using 10 seeds. All results are summarized in Table 11 and 12, from which we can find that FlipNet has the highest TAR and the lowest AFR in all cases. These results imply FlipNet has good action smoothness and noise robustness.

For comparing FlipNet and MLP-SN, we train them on DMControl Reacher environment. We use a 3-layer MLP-SN network and manually tuning its spectral norm of each layer by grid search. The

Table 9: **Environment-related hyperparameters in DMControl.**

| Env | $\lambda_k$ | $\lambda_h$ | Length of his. obsv. $N$ |
|---|---|---|---|
| Cartpole | $10^{-2}$ | $10^{-2}$ | 5 |
| Reacher | $10^{-2}$ | $10^{-3}$ | 5 |
| Cheetah | $10^{-3}$ | $10^{-3}$ | 5 |
| Walker | $10^{-2}$ | $10^{-3}$ | 10 |

global Lipschitz constant of MLP-SN is the product of the spectral norms of all layers. The results are listed in Table 13, from which we can find that FlipNet outperforms MLP-SN under all hyper-parameter setting. We refrain from comparing FlipNet to MLP-SN across all environments used in this paper, because this would necessitate the manual tuning of spectral norm hyperparameters for each layer, which have an unwieldy number of potential hyperparameter combinations.

Table 10: **Observation noise in DMControl.** The observation noise in each dimension is distributed in $U(-\sigma, \sigma)$.

| Env | Noise amplitude $\sigma$ |
|---|---|
| Cartpole | $[0.1, 0.1, 0.1, 0.2, 0.2]$ |
| Reacher | $[0.001 \text{ repeats } 7 \text{ times}]$ |
| Cheetah | $[0.01, 0.01, 0.05, 0.05, 0.05, 0.05, 0.05, 0.05,$ $0.5, 0.05, 0.1, 0.5, 0.5, 0.5, 0.5, 0.5, 0.5]$ |
| Walker | $[0.25 \text{ repeats } 24 \text{ times}]$ |

Table 11: **Total average return in DMControl.**

| Environment | | MLP | LipsNet-G | LipsNet-L | FlipNet |
|---|---|---|---|---|---|
| noise-free env | Cartpole | $805_{\pm 0.8}$ | $691_{\pm 1.0}$ | $831_{\pm 0.9}$ | $\mathbf{841}_{\pm 0.2}$ |
| | Reacher | $981_{\pm 10}$ | $979_{\pm 11}$ | $983_{\pm 10}$ | $\mathbf{988}_{\pm 10}$ |
| | Cheetah | $816_{\pm 30}$ | $702_{\pm 10}$ | $822_{\pm 4}$ | $\mathbf{829}_{\pm 15}$ |
| | Walker | $926_{\pm 12}$ | $956_{\pm 20}$ | $945_{\pm 13}$ | $\mathbf{962}_{\pm 10}$ |
| noisy env | Cartpole | $763_{\pm 9}$ | $517_{\pm 41}$ | $823_{\pm 6}$ | $\mathbf{825}_{\pm 3}$ |
| | Reacher | $972_{\pm 25}$ | $973_{\pm 18}$ | $978_{\pm 17}$ | $\mathbf{982}_{\pm 10}$ |
| | Cheetah | $813_{\pm 29}$ | $680_{\pm 7}$ | $818_{\pm 11}$ | $\mathbf{822}_{\pm 11}$ |
| | Walker | $911_{\pm 26}$ | $942_{\pm 15}$ | $929_{\pm 11}$ | $\mathbf{961}_{\pm 12}$ |

Additionally, the learned filter matrix $H$ in FlipNet is visualized in Figure 24 to show the noise filtering ability. Figure 24(a) and 24(b) show the frequency distributions of observation in noise-free environment and noisy environment, respectively. Their shades of color represent the intensity of frequency. The color in Figure 24(c) denotes the magnitude of elements in matrix $H$, which determines which frequencies are suppressed or strengthened. The result implies that the learned filter matrix mainly focus on the frequencies that containing observation information, and rarely focus on the frequencies that containing noises. In other words, FlipNet can automatically extract the important frequencies and filter out the noise frequencies.

Table 12: **Action fluctuation ratio in DMControl.**

| Environment | | MLP | LipsNet-G | LipsNet-L | FlipNet |
|---|---|---|---|---|---|
| noise-free env | Cartpole | $0.04 \pm 0.00$ | $0.08 \pm 0.00$ | $0.01 \pm 0.00$ | $\mathbf{0.01} \pm 0.00$ |
| | Reacher | $2.07 \pm 0.60$ | $0.13 \pm 0.24$ | $0.01 \pm 0.00$ | $\mathbf{0.01} \pm 0.00$ |
| | Cheetah | $1.08 \pm 0.02$ | $0.92 \pm 0.01$ | $0.94 \pm 0.01$ | $\mathbf{0.90} \pm 0.01$ |
| | Walker | $1.89 \pm 0.02$ | $1.25 \pm 0.02$ | $0.93 \pm 0.01$ | $\mathbf{0.74} \pm 0.01$ |
| noisy env | Cartpole | $0.58 \pm 0.03$ | $0.75 \pm 0.09$ | $0.17 \pm 0.01$ | $\mathbf{0.13} \pm 0.00$ |
| | Reacher | $2.41 \pm 0.28$ | $0.04 \pm 0.00$ | $0.04 \pm 0.03$ | $\mathbf{0.01} \pm 0.00$ |
| | Cheetah | $1.13 \pm 0.02$ | $1.00 \pm 0.01$ | $1.08 \pm 0.01$ | $\mathbf{0.94} \pm 0.01$ |
| | Walker | $2.02 \pm 0.03$ | $1.68 \pm 0.01$ | $1.21 \pm 0.01$ | $\mathbf{0.78} \pm 0.01$ |

Table 13: **Performance of FlipNet and MLP-SN on DMControl Reacher.**

| Name | Network Spectral norm for each layer | Total average return | Action fluctuation ratio |
|---|---|---|---|
| MLP-SN | 5.0 | $760 \pm 381$ | $0.01 \pm 0.00$ |
| | 5.5 | $831 \pm 102$ | $0.01 \pm 0.00$ |
| | 5.8 | $954 \pm 10$ | $0.08 \pm 0.05$ |
| | 6.0 | $967 \pm 28$ | $0.13 \pm 0.08$ |
| FlipNet | | $\mathbf{988} \pm 10$ | $\mathbf{0.01} \pm 0.00$ |

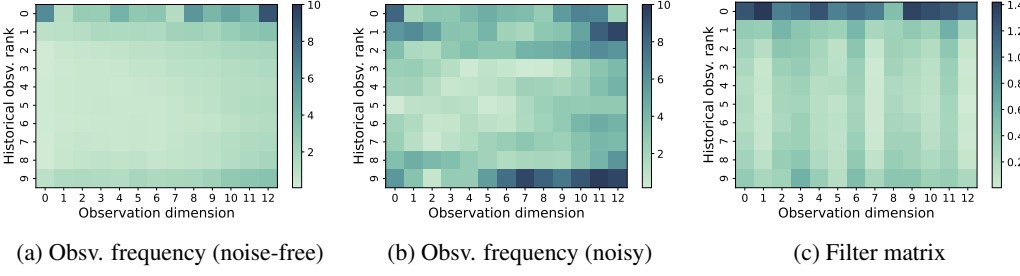

(a) Obsv. frequency (noise-free)  (b) Obsv. frequency (noisy)  (c) Filter matrix

Figure 24: **Filter matrix and observation frequency in walker environment.** The color in (a) and (b) represents the intensity of frequency. The color in (c) represents the magnitude of elements in matrix $H$. The color distribution in (c) implies FlipNet can automatically extract the important frequencies and filter out the noise frequencies.

## K COMPARISON TO REWARD PENALTY

Punishing the difference between consecutive actions in the reward is an effective way to smooth the actions in some environments. However, such an approach breaks the Markov property, which affects the performance, albeit to a minor extent in certain environments. Moreover, we found that adding reward penalty in a sparse reward environment increases action fluctuation rather than smoothing it, which is consistent with the finding by Chen et al. (2021) and Song et al. (2023).

Cartpole in DMControl is a sparse reward environment. The reward is 1 when the pole is within $30°$ of the vertical and 0 otherwise. We implement TD3 in this environment, punishing the difference between consecutive actions in the reward. Specifically, the new reward is $r = r_{\text{origin}} + \alpha \|a_{t+1} - a_t\|$, where $r_{\text{origin}}$ is the original sparse reward, $\alpha$ is the penalty coefficient and $a_{t+1}$ is the output of actor network under $s_{t+1}$. The experiment results are summarized in Table 14. The results imply that simply adding reward penalty in the sparse reward environment increases the action fluctuation ratio. Superiorly, FlipNet can smooth actions even in the sparse reward environment.

Table 14: Comparison to reward penalty.

| Method | Penalty coefficient $\alpha$ | Total average return | Action fluctuation ratio |
|---|---|---|---|
| TD3 (MLP, reward penalty) | 0.01 | 825 $\pm$ 0.5 | 0.27 $\pm$ 0.01 |
| TD3 (MLP, reward penalty) | 0.1 | 819 $\pm$ 0.8 | 0.21 $\pm$ 0.01 |
| TD3 (MLP, reward penalty) | 1 | 13 $\pm$ 0.5 | 0.02 $\pm$ 0.00 |
| TD3 (MLP) | | 805 $\pm$ 0.8 | 0.04 $\pm$ 0.00 |
| TD3 (LipsNet-G) | | 691 $\pm$ 1.0 | 0.08 $\pm$ 0.00 |
| TD3 (LipsNet-L) | | 831 $\pm$ 0.9 | 0.01 $\pm$ 0.00 |
| TD3 (FlipNet) | | **841** $\pm$ 0.2 | **0.01** $\pm$ 0.00 |

## L MINI-VEHICLE DRIVING: INTRODUCTION OF VEHICLE AND TASK

The vehicle robot is driven by two differential wheels, which is shown in Figure 26. The task for the robot is to track a given reference trajectory and reference velocity while avoiding obstacle. The setting of observations and actions in this environment is described in Table 15.

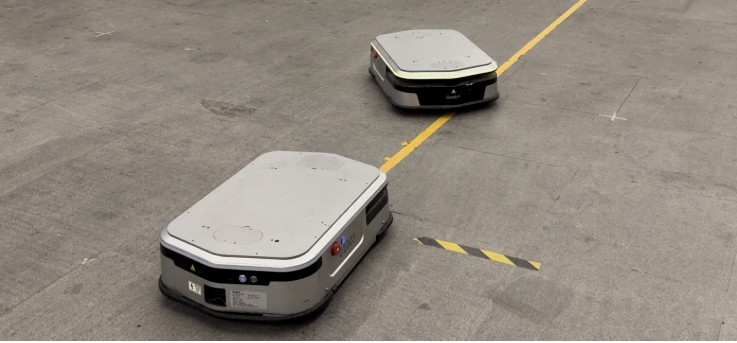

Figure 25: **Physical vehicle robots.**

For the perception, the vehicle is equipped with LiDAR, obtaining its position by matching with a pre-scanned point cloud map generated by SLAM. In this way, vehicle can detect its horizontal coordinate $x$, vertical coordinate $y$, and heading angle $\phi$. The vehicle is also equipped with a speed sensor that measures the linear velocity $v$ and angular velocity $\omega$. To increase the complexity of the task, another vehicle is used as a obstacle vehicle. Both vehicles can exchange real-time state information with each other via WiFi communication.

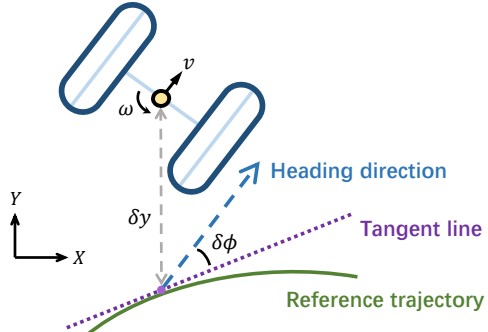

Figure 26: **Vehicle kinematics model.** The vehicle moves by two differential wheels, tracking the reference trajectory.

Table 15: **Variables in mini-vehicle driving env.**

| Variable | | Description |
|---|---|---|
| | $v$ | longitudinal speed |
| | $\omega$ | yaw rate |
| | $\delta y$ | trajectory offset |
| | $\delta \phi$ | heading angle error |
| Obsv. | $\delta v$ | speed error |
| | $\Delta x$ | obstacle's relative $x$ position |
| | $\Delta y$ | obstacle's relative $y$ position |
| | $\Delta \phi$ | obstacle's relative angle |
| | $\Delta v$ | obstacle's relative speed |
| | $\Delta \omega$ | obstacle's relative yaw rate |
| Action | $\dot{v}$ | longitudinal acceleration |
| | $\dot{\omega}$ | yaw acceleration |

For the decision-making and control, a policy network trained by RL is deployed on the vehicle. After inputting the perceived observation into the network, control actions are computed, namely linear acceleration $\dot{v}$ and angular acceleration $\dot{\omega}$. Then, control actions are sent to the motor to execute the command. The overall control mode is shown in Figure 27.

As illustrated in Section 4.3, there are four diverse scenarios in this environment. The scenario descriptions are listed in Table 1. To describe the scenario settings more clearly, Figure 28 shows the map and vehicle routes for each scenario. Figure 29 shows the corresponding snapshot for each scenario. In scenarios 1-3, the obstacle vehicle goes straight with constant speed. In scenario 4, the obstacle vehicle is manipulated by human.

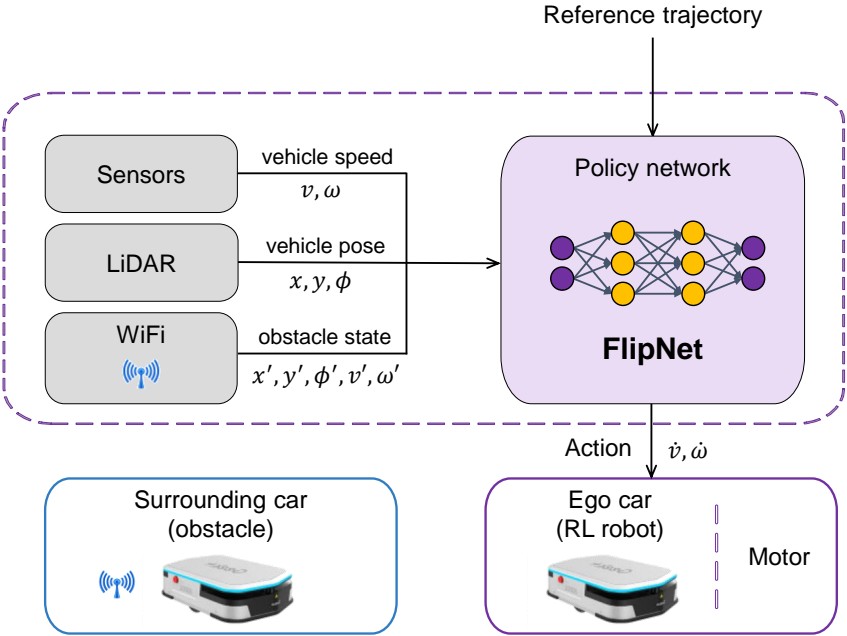

Figure 27: **Flowchart of vehicle control mode.**

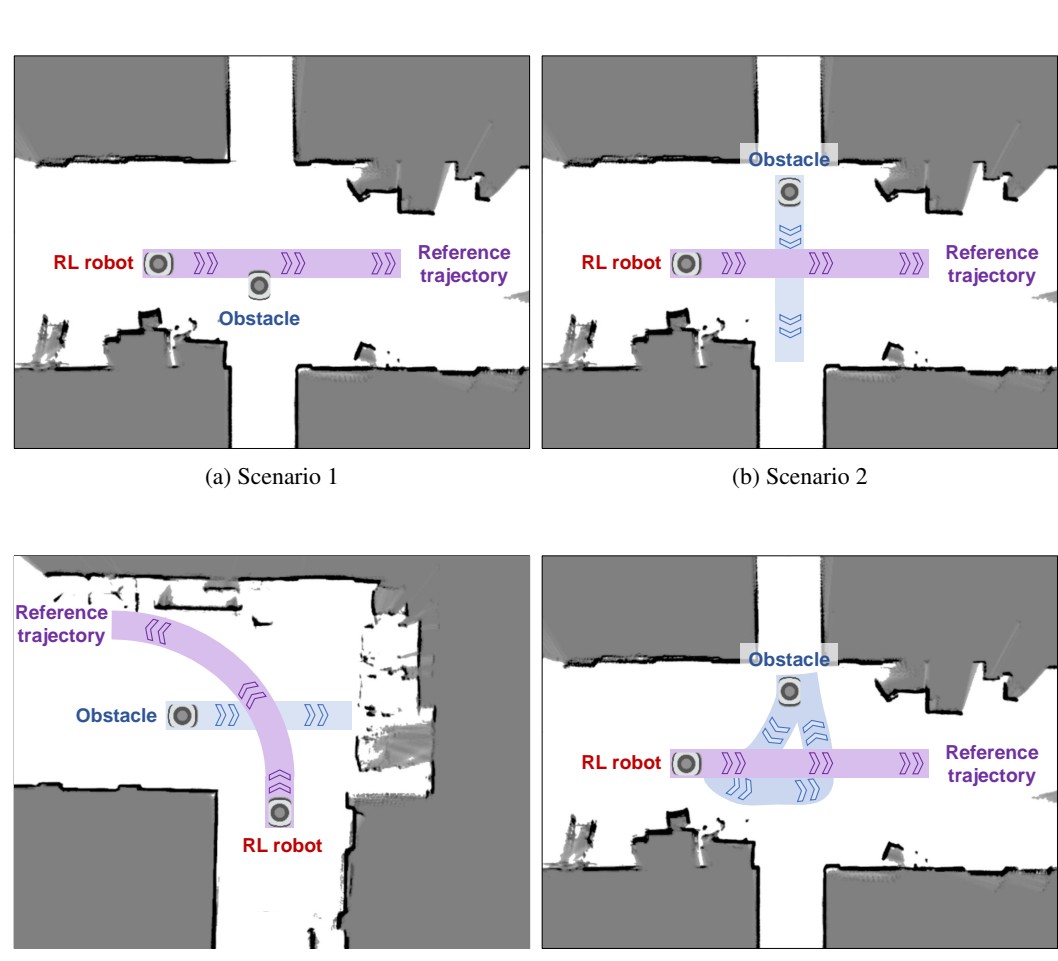

(a) Scenario 1

(b) Scenario 2

(c) Scenario 3

(d) Scenario 4

Figure 28: **Scenario illustration of mini-vehicle driving environment.**

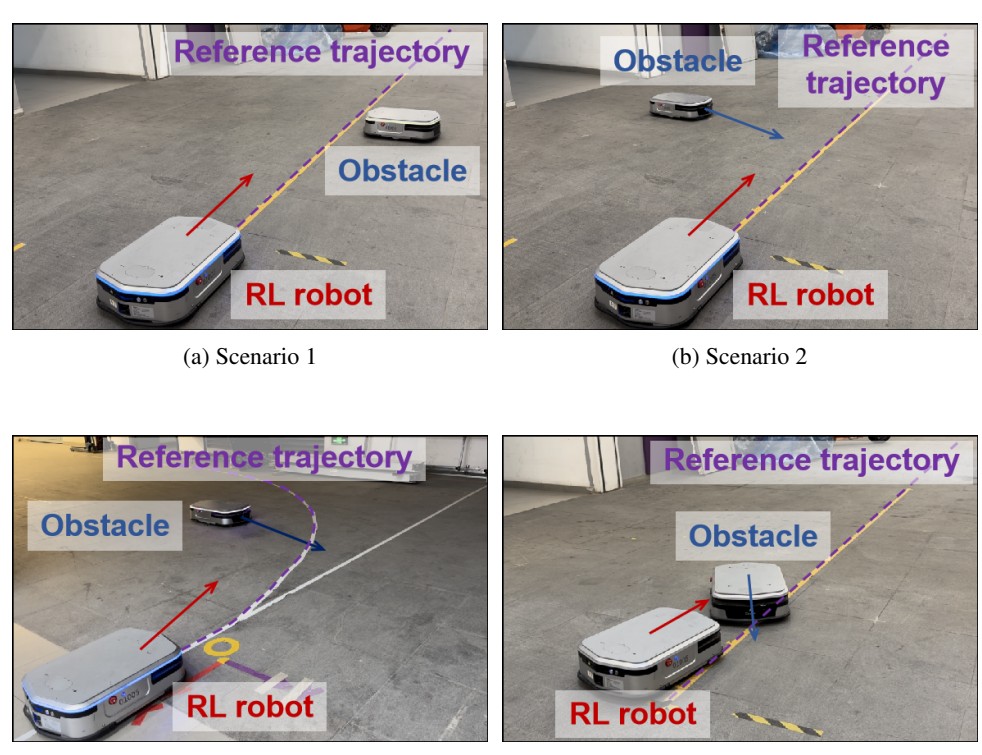

(a) Scenario 1                    (b) Scenario 2

(c) Scenario 3                    (d) Scenario 4

Figure 29: **Scenario snapshots of mini-vehicle driving environment.**

## M    MINI-VEHICLE DRIVING: DETAILED IMPLEMENTATION AND RESULTS

In the training stage, observation noise is set to zero. In the vehicle testing stage, multiple different magnitudes of observation noise are added to thoroughly test the performance of policy networks. The noise magnitude is adjusted using the coefficient $\sigma_{\text{coef}} \in \mathbb{R}^+ \cup \{0\}$, such that noise is distributed in $U(\sigma_{\text{coef}} \cdot \sigma_{\text{base}})$. And the baseline noise $\sigma_{\text{base}}$ is set to:

$$\sigma_{\text{base}} = \begin{bmatrix} 0.01 & \frac{\pi}{180} & 0.03 & \frac{\pi}{180} & 0.01 & 0.03 & 0.03 & \frac{\pi}{180} & 0.01 & \frac{\pi}{180} \end{bmatrix}^\top.$$

The reward function is defined as a constant minus the penalties related to tracking error, vehicle instability, and collision violation:

$$r = 1 - 0.4(\delta y)^2 - 0.1(\delta \phi)^2 - 1.3|\delta v| - 0.01\omega^2 - 0.01\dot{v}^2 - 0.01\dot{\omega}^2 - 2 \cdot \mathbb{I}(\rho < 0.94),$$

where $\rho$ represents the distance between the centers of the two vehicles, calculated as $\rho = \sqrt{\Delta x^2 + \Delta y^2}$. The reference speed is set to 0.3m/s, meaning $\delta v = v - 0.3$.

The Distributional Soft Actor-critic (DSAC) (Duan et al., 2021), a model-free RL algorithm, is used to train the vehicle robot. The hyperparameters for DSAC are listed in Table 16. The tests in all scenarios are accomplished by the same networks.

All results are shown in Figure 30∼48. Table 17 lists the figure index for each scenario.

Table 16: **Hyperparameters for DSAC.**

| Hyperparameter | Value |
|---|---|
| Replay buffer capacity | 1000000 |
| Buffer warm-up size | 10000 |
| Batch size | 256 |
| Discount $\gamma$ | 0.99 |
| Target network soft-update rate $\tau$ | 0.005 |
| Policy delay times | 2 |
| Temperature parameter $\alpha$ | 0.2 |
| Hidden layers in critic network | [256, 256] |
| Activations in critic network | ReLU |
| Hidden layers in subnetwork $f$ | [256, 256] |
| Activations in subnetwork $f$ | ReLU |
| Optimizer | Adam |
| Critic learning rate | $1 \cdot 10^{-4}$ |
| Actor learning rate | $1 \cdot 10^{-4}$ |
| Coefficient $\lambda_k$ | 0.1 |
| Coefficient $\lambda_h$ | 0.04 |
| Length of historical obsv. $N$ | 20 |

Table 17: **Figure indices for the results of mini-vehicle driving environment.**

| Scenario and network | | Noise amplitude | | Snapshots |
|---|---|---|---|---|
| | | 0 | 10 | |
| Scenario 1 | MLP | Figure 30 | Figure 32 | Figure 34 |
| | FlipNet | Figure 31 | Figure 33 | |
| Scenario 2 | MLP | Figure 35 | Figure 37 | Figure 39 |
| | FlipNet | Figure 36 | Figure 38 | |
| Scenario 3 | MLP | Figure 40 | Figure 42 | Figure 44 |
| | FlipNet | Figure 41 | Figure 43 | |
| Scenario 4 | MLP | Figure 45 | Figure 47 | URL [5] |
| | FlipNet | Figure 46 | Figure 48 | |

---

[5] Project page: https://iclr-anonymous-2025.github.io/FlipNet

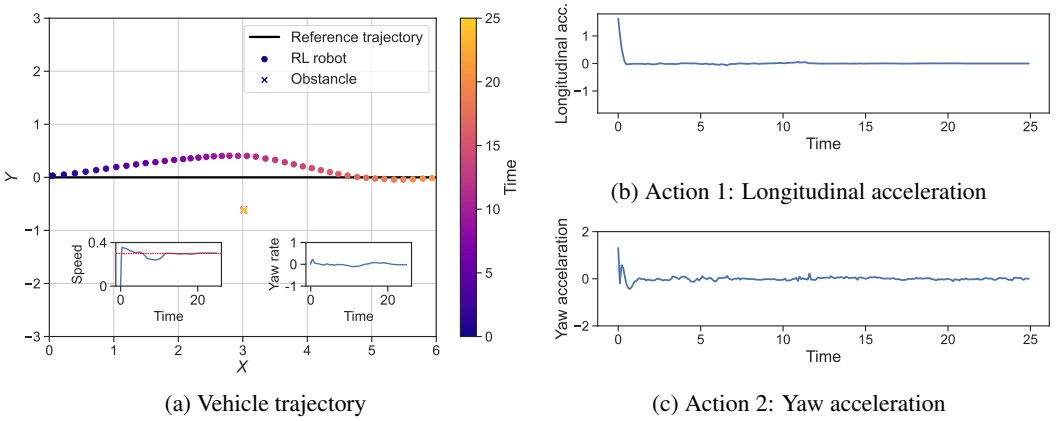

(a) Vehicle trajectory

(b) Action 1: Longitudinal acceleration

(c) Action 2: Yaw acceleration

Figure 30: **MLP performance in scenario 1.** The noise amplitude is 0.

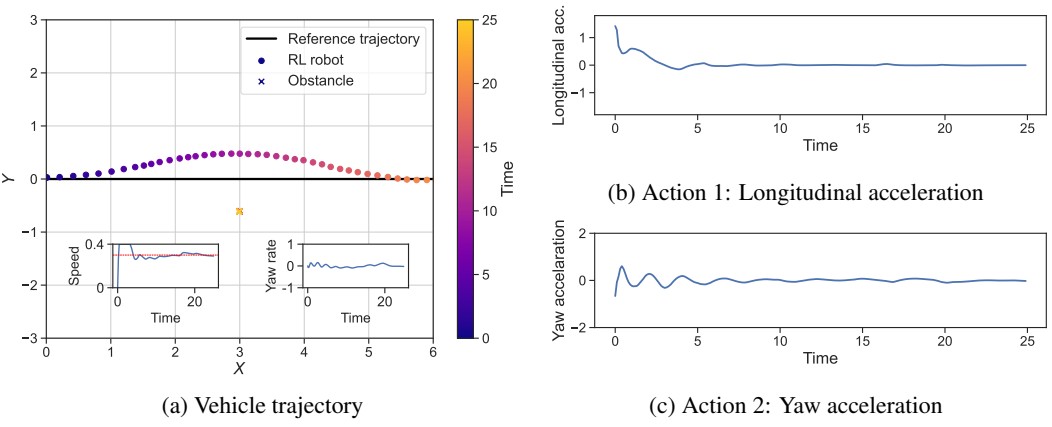

(a) Vehicle trajectory

(b) Action 1: Longitudinal acceleration

(c) Action 2: Yaw acceleration

Figure 31: **FlipNet performance in scenario 1.** The noise amplitude is 0.

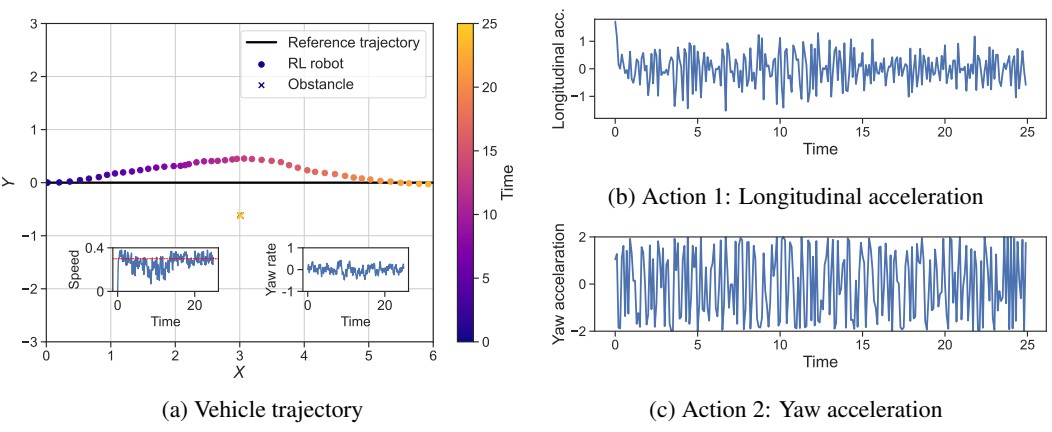

(a) Vehicle trajectory

(b) Action 1: Longitudinal acceleration

(c) Action 2: Yaw acceleration

Figure 32: **MLP performance in scenario 1.** The noise amplitude is 10.

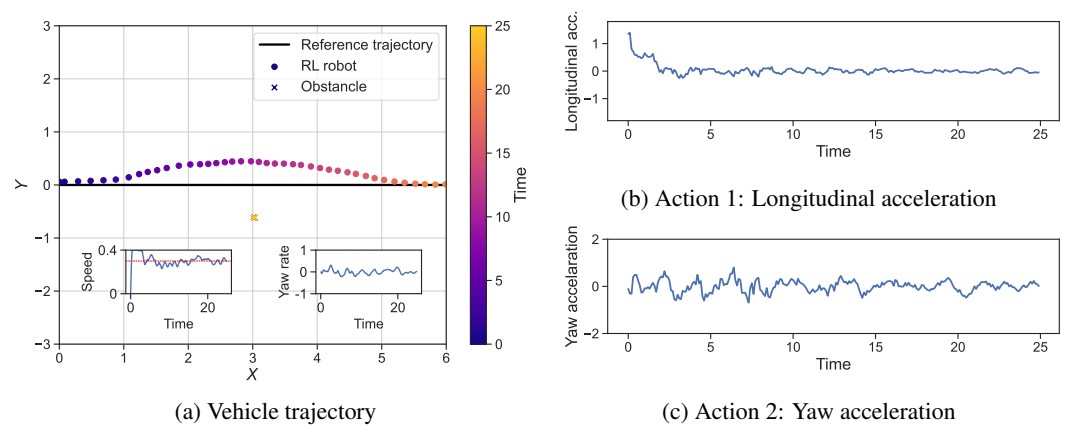

(a) Vehicle trajectory

(b) Action 1: Longitudinal acceleration

(c) Action 2: Yaw acceleration

Figure 33: **FlipNet performance in scenario 1.** The noise amplitude is 10.

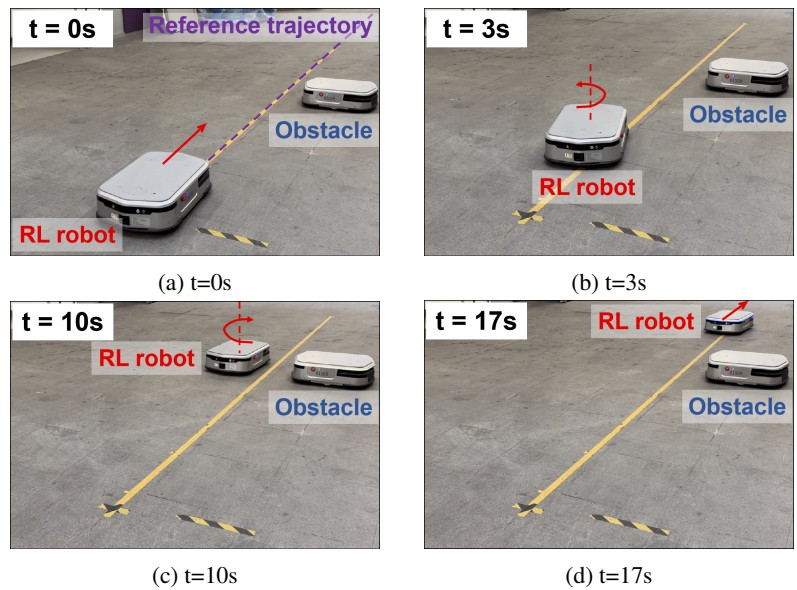

(a) t=0s

(b) t=3s

(c) t=10s

(d) t=17s

Figure 34: **Snapshots of scenario 1.**

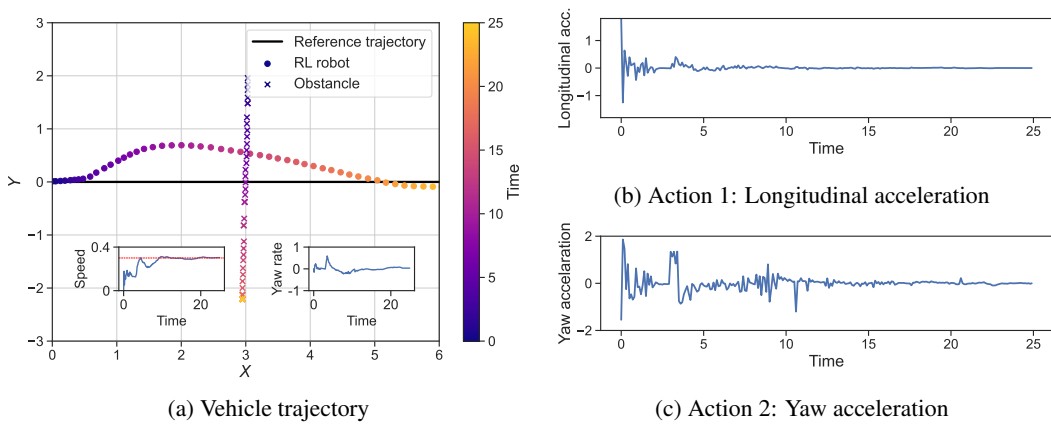

(a) Vehicle trajectory

(b) Action 1: Longitudinal acceleration

(c) Action 2: Yaw acceleration

Figure 35: **MLP performance in scenario 2.** The noise amplitude is 0.

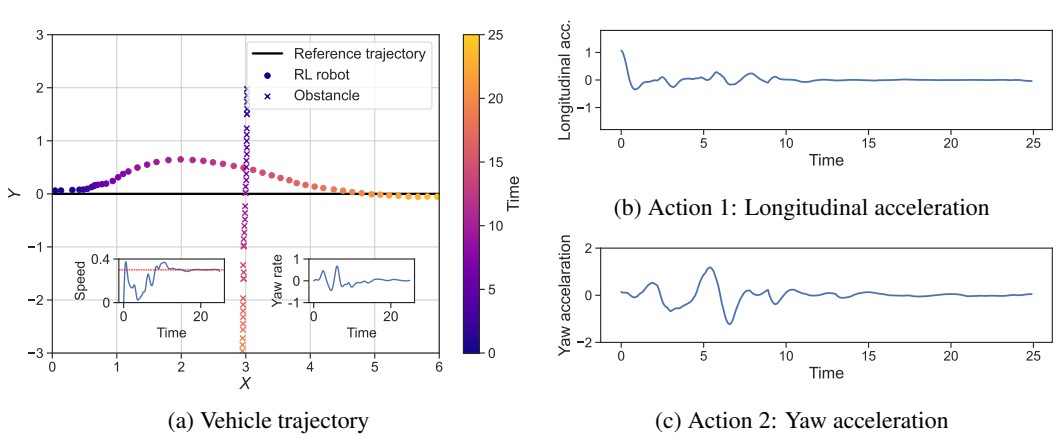

(a) Vehicle trajectory

(b) Action 1: Longitudinal acceleration

(c) Action 2: Yaw acceleration

Figure 36: **FlipNet performance in scenario 2.** The noise amplitude is 0.

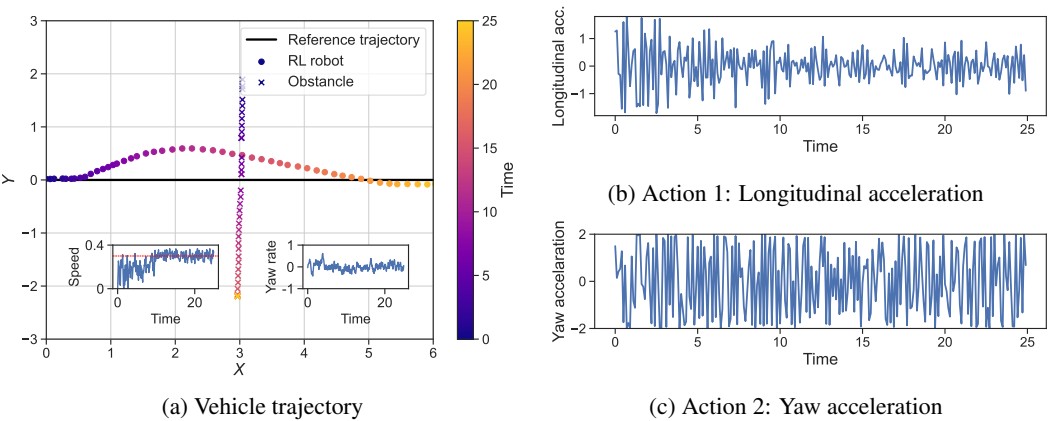

(a) Vehicle trajectory

(b) Action 1: Longitudinal acceleration

(c) Action 2: Yaw acceleration

Figure 37: **MLP performance in scenario 2.** The noise amplitude is 10.

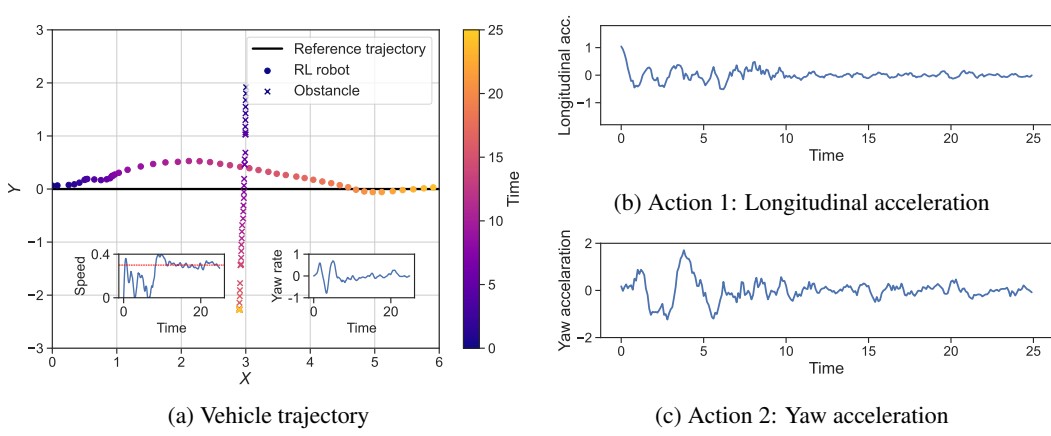

(a) Vehicle trajectory

(b) Action 1: Longitudinal acceleration

(c) Action 2: Yaw acceleration

Figure 38: **FlipNet performance in scenario 2.** The noise amplitude is 10.

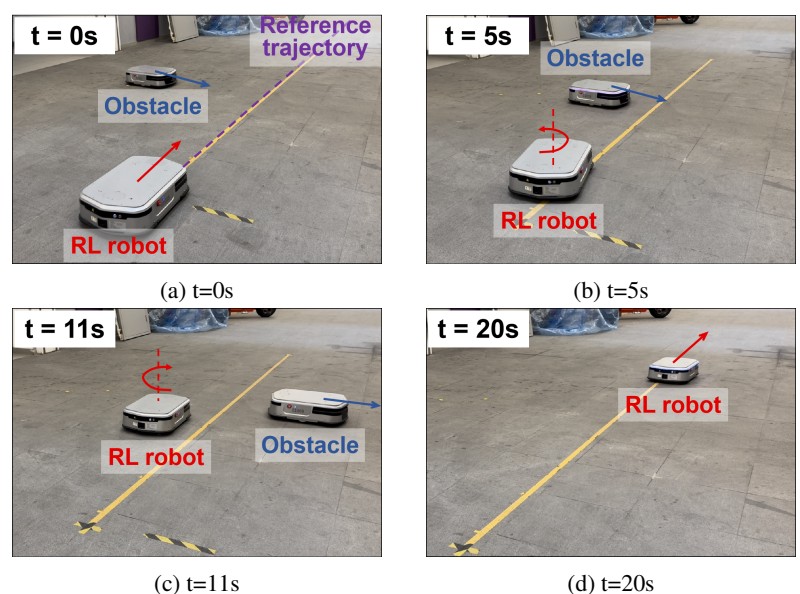

Figure 39: **Snapshots of scenario 2.**

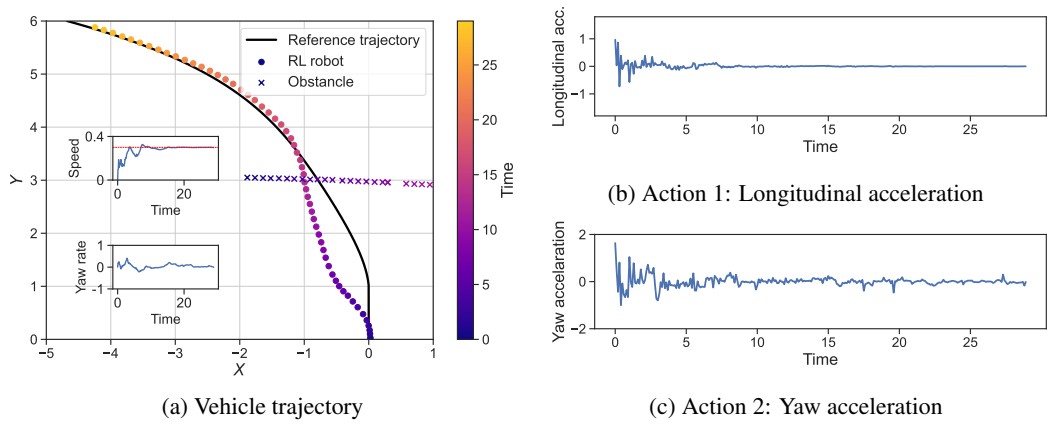

Figure 40: **MLP performance in scenario 3.** The noise amplitude is 0.

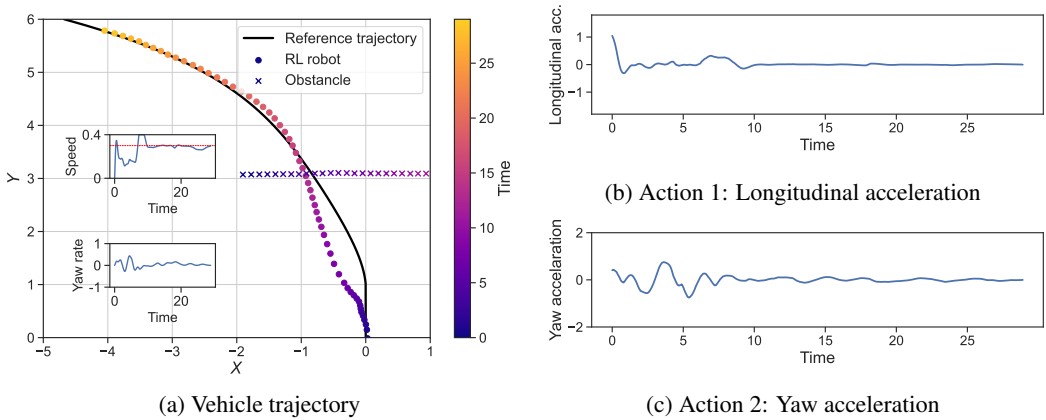

Figure 41: **FlipNet performance in scenario 3.** The noise amplitude is 0.

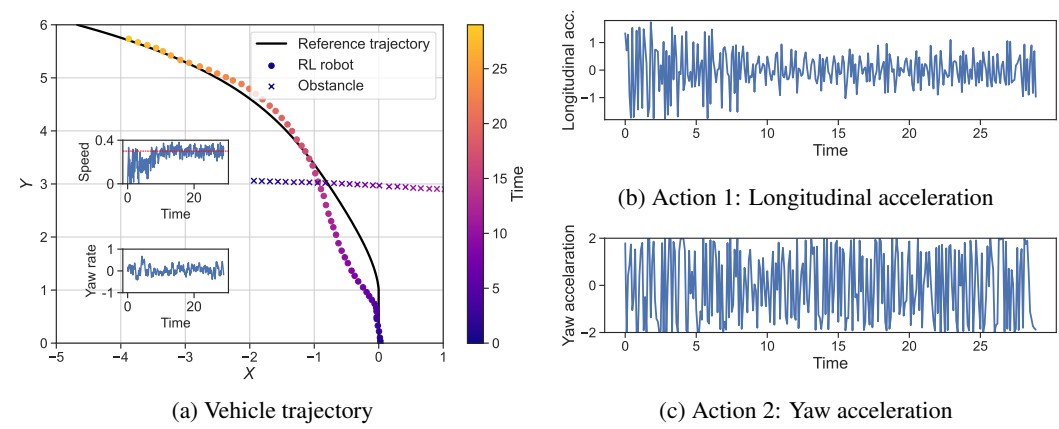

(a) Vehicle trajectory

(b) Action 1: Longitudinal acceleration

(c) Action 2: Yaw acceleration

Figure 42: **MLP performance in scenario 3.** The noise amplitude is 10.

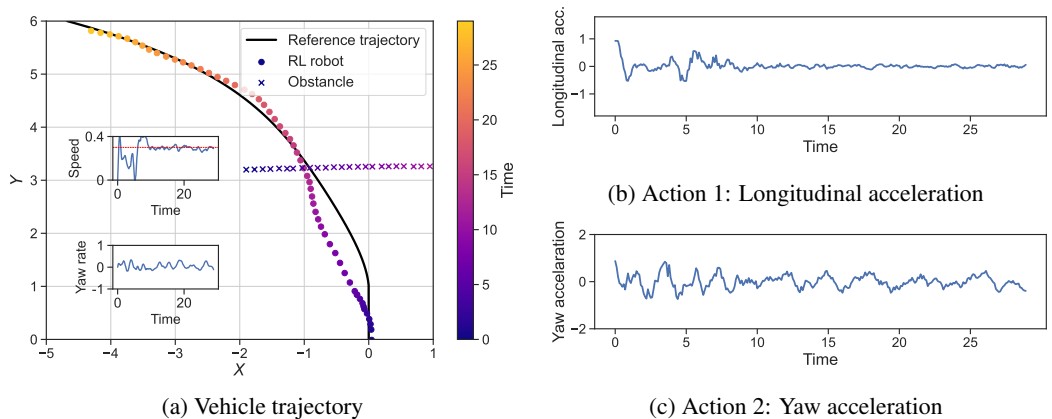

(a) Vehicle trajectory

(b) Action 1: Longitudinal acceleration

(c) Action 2: Yaw acceleration

Figure 43: **FlipNet performance in scenario 3.** The noise amplitude is 10.

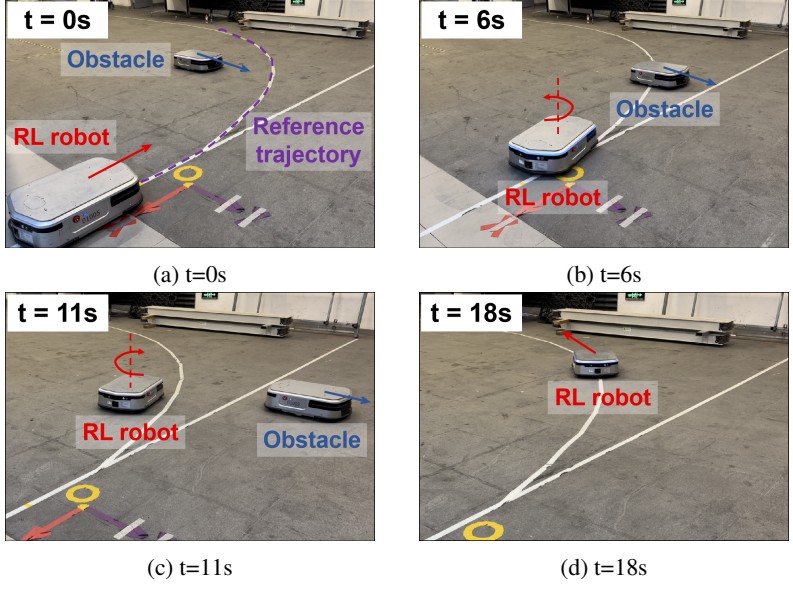

(a) t=0s

(b) t=6s

(c) t=11s

(d) t=18s

Figure 44: **Snapshots of scenario 3.**

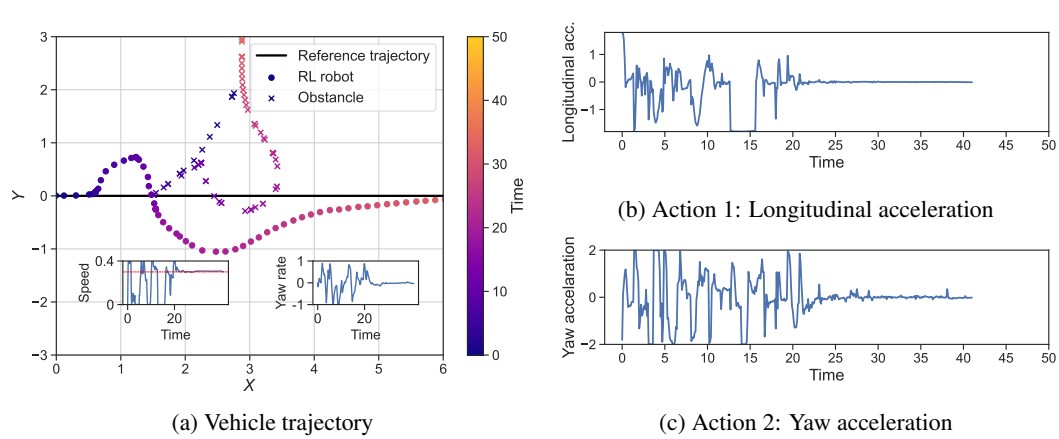

(a) Vehicle trajectory

(b) Action 1: Longitudinal acceleration

(c) Action 2: Yaw acceleration

Figure 45: **MLP performance in scenario 4.** The noise amplitude is 0.

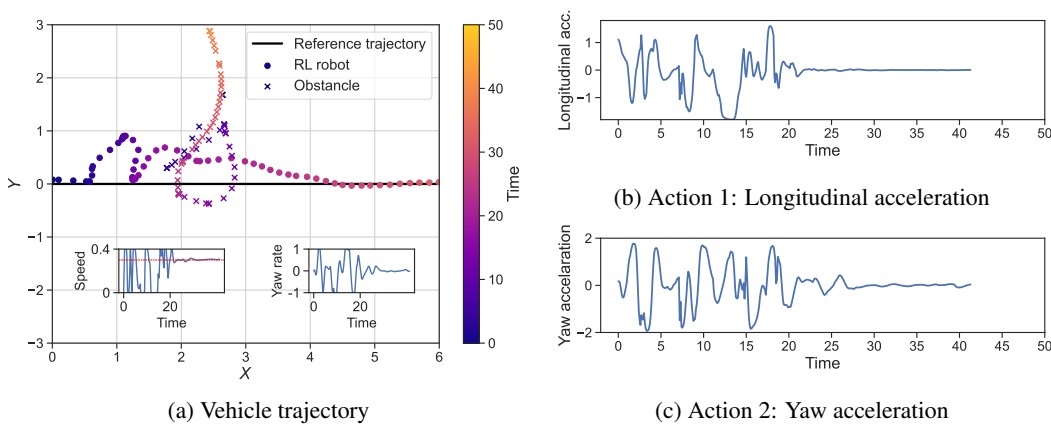

(a) Vehicle trajectory

(b) Action 1: Longitudinal acceleration

(c) Action 2: Yaw acceleration

Figure 46: **FlipNet performance in scenario 4.** The noise amplitude is 0.

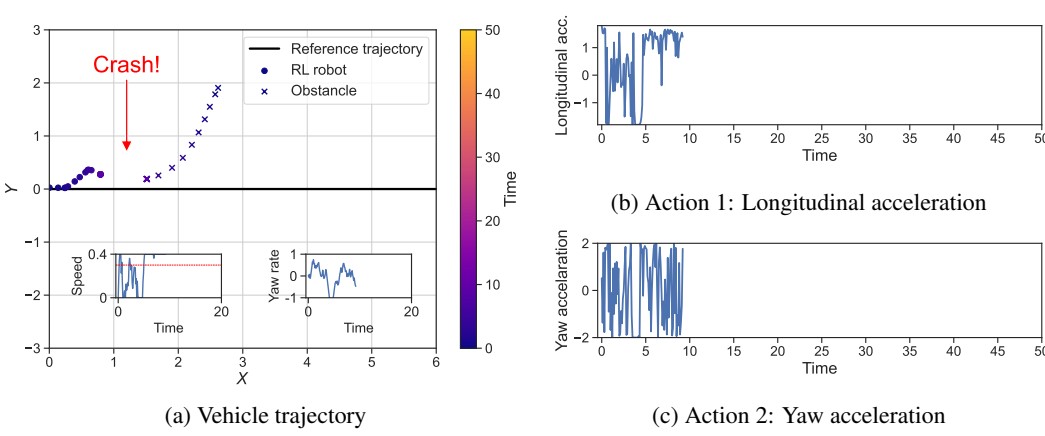

(a) Vehicle trajectory

(b) Action 1: Longitudinal acceleration

(c) Action 2: Yaw acceleration

Figure 47: **MLP performance in scenario 4.** The noise amplitude is 10.

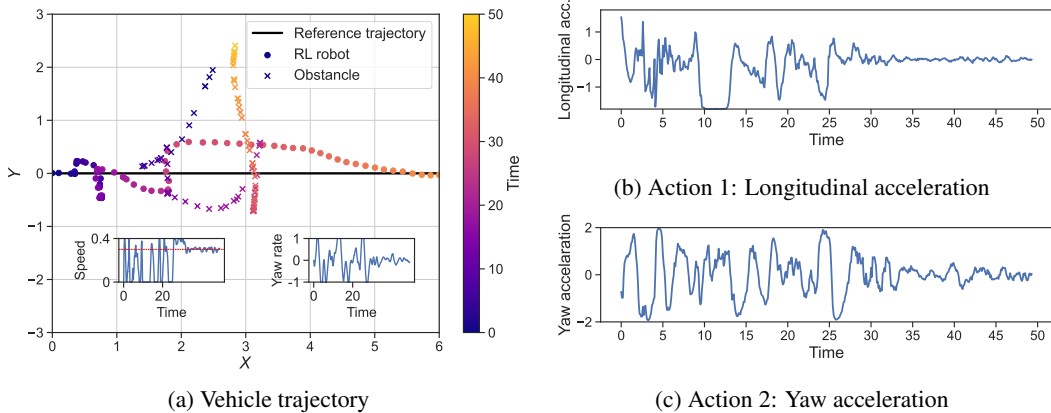

(a) Vehicle trajectory

(b) Action 1: Longitudinal acceleration

(c) Action 2: Yaw acceleration

Figure 48: **FlipNet performance in scenario 4.** The noise amplitude is 10.

The results of TAR and AFR for scenario 1-3 are listed in Table 18. The result for scenario 4 is not listed because the obstacle vehicle is manipulated by human, which means each trial has great randomness. The data in Table 18 is visualized in Figure 49. As shown in Figure 49(a)(c)(e), when noise increases, FlipNet maintains the highest TAR and its TAR declines much slower than MLP's. As shown in Figure 49(b)(d)(f), when noise increases, FlipNet maintains the lowest AFR and its AFR grows much slower than MLP's. These results imply FlipNet has excellent action smoothness and noise robustness.

Table 18: **Performance summary in mini-vehicle driving environment.**

| Task setting | | Scenario 1 | | Scenario 2 | | Scenario 3 | |
|---|---|---|---|---|---|---|---|
| Policy network | Noise amplitude | TAR | AFR | TAR | AFR | TAR | AFR |
| FlipNet | 0 | 234.7 | **0.02** | 252.6 | **0.04** | 287.5 | **0.03** |
| | 1 | 235.2 | **0.02** | **252.0** | **0.04** | 288.5 | **0.03** |
| | 5 | **232.8** | **0.08** | **254.1** | **0.08** | **289.6** | **0.08** |
| | 10 | **233.6** | **0.14** | **249.6** | **0.16** | **290.3** | **0.14** |
| | 20 | **224.5** | **0.27** | **252.7** | **0.28** | **281.3** | **0.23** |
| MLP | 0 | **238.4** | 0.04 | **254.6** | 0.17 | **293.5** | 0.15 |
| | 1 | **237.8** | 0.58 | 250.4 | 0.58 | **293.0** | 0.55 |
| | 5 | 232.7 | 1.68 | 250.0 | 1.62 | 289.6 | 1.58 |
| | 10 | 225.0 | 2.03 | 247.2 | 2.24 | 283.3 | 2.17 |
| | 20 | 209.8 | 2.53 | 238.9 | 2.65 | 267.9 | 2.65 |

## N    LIMITATIONS, FUTURE WORKS, AND COMMUNITY IMPACTS

FlipNet achieves smoother and more robust control with a slight increase in training time, as shown in Appendix H. In the future, we plan to optimize the backward time of FlipNet. We have devised a solution to accelerate it by using multiple forward propagation and zero-order gradient estimation to compute the Jacobian matrix. Furthermore, we plan to introduce an attention mechanism for the filter matrix $H$ in the future works. In this way, $H$ can vary according to different observation inputs. Additionally, We are now trying to implement FlipNet on a real-world highway vehicle. Complete results of all the above future improvements will be soon reported in our next work.

As for the positive impacts on the AI community, FlipNet addresses the action fluctuation problem of RL. FlipNet breaks through the bottleneck of action fluctuation and poor robustness faced by RL, which accelerates the process of RL's real-world application. It mitigates the wear of actuators, safety risks, and performance reduction caused by action fluctuation. FlipNet benefits many industrial fields, including robot control, drone control, decision-making and control of autonomous vehicles, and embodied AI.

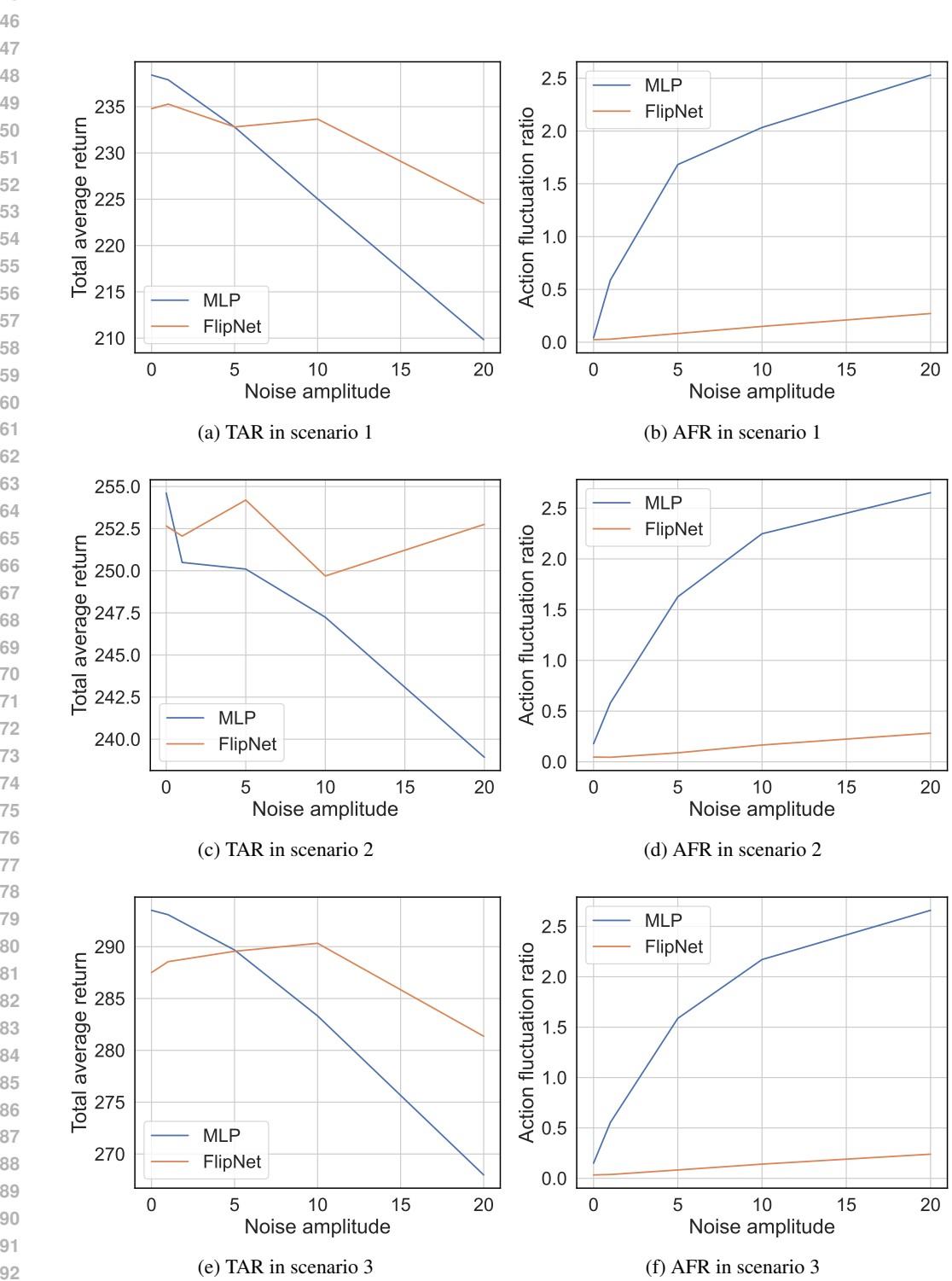

(a) TAR in scenario 1

(b) AFR in scenario 1

(c) TAR in scenario 2

(d) AFR in scenario 2

(e) TAR in scenario 3

(f) AFR in scenario 3

Figure 49: **Performance trend with increasing noise in mini-vehicle driving environment.**

