# OpenReview forum: "FlipNet: Fourier Lipschitz Smooth Policy Network for Reinforcement Learning"
_ICLR.cc/2025/Conference — Submitted to ICLR 2025_

### Official Review · Reviewer_du6n · 2024-10-29

**Soundness:** 2
**Presentation:** 3
**Contribution:** 3
**Rating:** 6
**Confidence:** 4

**Summary:**

This paper proposes a new policy network architecture called Fourier Lipschitz Smooth Policy Network (FlipNet), with the aim to address the action fluctuation problem in deep RL. Through identifying  With a continuous-time analysis, two causes of action fluctuation are identified, i.e., policy network smoothness and observation noise. FlipNet then adopts two techniques, Jacobian regularization and Fourier filter layer, to deal with the two causes respectively. The proposed method is evaluated in a classic linear quadratic control task, four DMC tasks, and four Mini-vehicle driving tasks, showing a great reduction of action fluctuation with a slight score improvement at the same time.

**Strengths:**

- The paper is well written and organized. Illustrations are well used to describe the ideas and methods.
- The motivation is almost clear. The transition to the proposal of the two techniques is smooth. The differences between the proposed method and the previous methods are presented clearly.
- Three types of environments are used for evaluation. Other results for the ablation study, and hyperparameter analysis are provided in the appendix. I recommend the authors adjust the content and move the ablation study from the appendix to the main body, because I believe it is important for the audience to understand the effect of the proposed method.

**Weaknesses:**

- The motivative analysis in Equation 5 is good. However, taking the second term at the right hand side of the equation as the observation noise needs more discussion and justification. A natural question here is, whether the second term should be related to the transition dynamics (as well as the actions selected by policy). This concern is not well discussed and addressed in Section 3.1.
- As the observation noise is one of the main courses identified in this work. More discussion and analysis are needed. What are the noise models? In Section 4.1, a uniform distribution is used as the model of noise. For DMC, Table 9 presents the uniform distributions. And for Mini-vehicle driving, the noise model used is not clear in the main body and also in the appendix L.
- FlipNet introduces three new hyperparameters, i.e., $\lambda_k, \lambda_h$ and $N$. According to Table 6, it seems that the backward computation time of FlipNet is about 10 times of MLP. Therefore, it remains questionable whether the proposed method can be useful and feasible in other problems.
- Four tasks from DMC are a bit insufficient to evaluate the performance of the proposed method thoroughly. I suggest the authors add 4 more tasks, e.g., dog, quarduped, hopper, ant. The choice of noises for the noisy environments is not well discussed and justified (see my question below).
- Some experimental details are missing. Please refer to my questions below.

&nbsp;

### Minors

- In Line 51, “MLP-SN suffer” should be “MLP-SN suffers”.
- In Line 124, “Take DDPG as an example again” should be “Taking DDPG as an example again”.

**Questions:**

1. In Equation 5, is the second term at the right hand side of the equation also determined by transition dynamics?
2. $j$ is not defined in Equation 8. What is definition and meaning of it? Should it be $i$, i.e., the imaginary unit?
3. The trainable filter matrix $H$ depends on the dimensionality of the observation $D$. What if $D$ is high as commonly seen in real-world problems? Will it increase the training difficulty and cripple the performance of the proposed method? I would like to see more discussion or additional experiments on this point.
4. According to Table 6, it seems that the backward computation time of FlipNet is about 10 times of MLP. I would like to know the training time of FlipNet, as the back-propagation is different a lot from regular policy networks. A comparison regarding wall-clock training time between FlipNet and regular MLP will help a lot.
5. In Line 269, the authors mentioned “By choosing H as a complex matrix instead of real matrix, the Fourier filtering layer can not only alter frequency amplitudes but also perform feature extraction”. Can the authors provide more explanation for this?
6. How many seeds are used for Figure 9, Table 10, Table 11?
7. How are the observation noise amplitudes in Table 9 chosen?
8. Why are there no error bars in Figure 10 and Figure 13?

---

> ### Author Response · Authors · 2024-11-19
> **Rebuttal by Authors (1/2)**
>
> We thank you for the careful reading and insightful review!
>
> After carefully considering your concerns, we now address each of them below:
>
> ---
> #### **> Weakness 1 & Question 1**
> In RL control tasks, action is computed as $a_t = \pi(o_t)$ and observation is given by $o_t = s_t + \xi_t$, where $s_t$ represents the current state and $\xi_t$ denotes any type of observation noise.
> The term $\Vert \frac{\mathrm{d}o_t}{\mathrm{d}t} \Vert$ you mentioned is actually $\Vert \frac{\mathrm{d}o_t}{\mathrm{d}t} \Vert = \Vert \frac{\mathrm{d} (s_t + \xi_t)}{\mathrm{d}t} \Vert = \Vert \frac{\mathrm{d} s_t}{\mathrm{d}t} + \frac{\mathrm{d} \xi_t}{\mathrm{d}t} \Vert$. The first part $\frac{\mathrm{d} s_t}{\mathrm{d}t}$, i.e. the change rate of dynamics system, is related to the **inherent property** of the target dynamic system. Therefore, **when high-frequency noise occurs, the second part $\frac{\mathrm{d} \xi_t}{\mathrm{d}t}$ will causes $\Vert \frac{\mathrm{d}o_t}{\mathrm{d}t} \Vert$ to increase significantly.**
> Thank you for pointing out the potential source of misunderstanding. In the revised paper, we have decouple $o_t$ as two terms, i.e. $s_t$ and $\xi_t$, to make it clear. We kindly invite you to check Section 3.1 in the uploaded revision paper.
>
> #### **> Weakness 2**
> In fact, the noise model for Mini-Vehicle Driving **have already been illustrated in Appendix M of our original paper**. We kindly invite you to refer to the first paragraph in Appendix M, where both the noise amplitude and uniform distribution were clearly described.
> Additionally, we would like to emphasize that FlipNet does not require specific noise models. This is because the Fourier Filter Layer learns from the data whether each frequency should be suppressed or enhanced. **The learned optimal filter matrix depends on the training data rather than predefined noise models.**
>
> #### **> Weakness 3 & Question 4**
> (1) **For hyperparameters:**
> We have already done sensitivity analysis for $\lambda_k$, $\lambda_h$, and $N$ in Appendix F and G of our original paper.
> We kindly invite you to refer to Fig. 17, Fig. 18, and Fig. 19.
> The results show that all the hyperparameters have very low sensitivity,
> making FlipNet convenient for tuning and easy to use.
>
> (2) **For backward time:**
> We greatly thank you for poing out the difference between backward time and training wall-clock time.
> 10x backward time does not mean the training time is 10x slower. Backward propagation is just one step in RL. RL involves additional time-consuming steps beyond backward, such as sampling and evaluation. According to your suggestion, we list the training wall-clock times for 1M iterations of TD3 on DMControl envs below.
> | Network  | Cartpole | Reacher | Cheetah | Walker | Total  |
> |----------|----------|---------|---------|--------|--------|
> | MLP  | 120 min  | 118 min | 128 min | 125 min| 491 min|
> | FlipNet | 194 min  | 195 min | 206 min | 204 min| 799 min|
>
> On average, the training time of FlipNet is 1.6 times that of MLP, which more accurately illustrates the impact on training time. The result is updated in Appendix H. We kindly invite you to check it in the uploaded revision paper.
>
> In conclusion,
> the hyperparameters have low sensitivity and the training time is acceptable.
>
> #### **> Weakness 4**
> In addition to DMC, we also evaluate on the real-world Mini-Vehicle Driving task, which is a more valuable benchmark.
> We excluded DMC's Dog, Quadruped, Hopper, and Ant because TD3 performs poorly in these challenging environments, which fall outside the scope of our paper. Similarly, the baseline method, i.e. LipsNet, excluded these 4 environments in their paper for likely the same reason.
>
> In order to address your concern, we added 3 new environments in Gymnasium: Hopper, Ant, and Humanoid (Quadruped and Ant are the same environment in Gymnasium).
>
> Total average return:
> - |Env|Noise $\sigma$ | MLP | LipsNet | FlipNet |
> |---|---|---|---|---|
> |Hopper| 0.05 | 3304 ± 4 | 3461 ± 3 | 3460 ± 8 |
> || 0.1 | 3127 ± 388 | 3302 ± 21 | 3459 ± 15 |
> |Ant| 0.05 | 3610 ± 44 |  3791 ± 38 | 3804 ± 44 |
> || 0.1 | 1903 ± 206 | 2712 ± 301 | 3390 ± 449 |
> |Humanoid| 0.05 | 5021 ± 46 | 5267 ± 15 | 5503 ± 21 |
> || 0.1 | 4984 ± 39 | 5069 ± 22 | 5505 ± 13 |
>
> Action fluctuation ratio:
> - |Env|Noise $\sigma$ | MLP | LipsNet | FlipNet |
> |---|---|---|---|---|
> |Hopper| 0.05 | 1.00 ± 0.02 | 0.95 ± 0.01 | 0.90 ± 0.01 |
> || 0.1 | 1.22 ± 0.05 | 1.13 ± 0.03 | 1.01 ± 0.02 |
> |Ant| 0.05 | 1.30 ± 0.01 | 1.31 ± 0.01 | 1.30 ± 0.01 |
> || 0.1 | 1.56 ± 0.04 | 1.42 ± 0.02 | 1.35 ± 0.01 |
> |Humanoid| 0.05 | 1.00 ± 0.02 | 0.79 ± 0.02 | 0.67 ± 0.02 |
> || 0.1 | 1.07 ± 0.01 | 0.90 ± 0.02 | 0.72 ± 0.02 |
>
> In conclusion, FlipNet has significantly better robostness and smoothness in high noise situations.
>
> Continue in the next reponse ...

---

> ### Author Response · Authors · 2024-11-19
> **Rebuttal by Authors (2/2)**
>
> Continue here ...
>
> #### **> Minors**
> All typos have been corrected in the uploaded revision paper.
>
> #### **> Question 2**
> Yes, $j$ is the imaginary unit. Its definition has been added in the uploaded revision paper.
>
> #### **> Question 3**
> FlipNet performs well in high-dimensional observation tasks.
> The newly added Humanoid environment is such a task with **348 observations**.
> And the above results show that FlipNet still maintains good robostness and smoothness.
>
> Additionally, the FFL (Fourier Filter Layer) is an network layer, which can be inserted in arbitrary position beyond the input layer in the paper.
> This means you can **first use a layer for feature dimensionality reduction and then pass the features through the FFL**.
> We added an experiment using Humanoid where the policy network is [Linear(348, 256), ReLU, FFL, Linear(256, 256), ReLU, Linear(256, 17)]. The following results show that the network with FFL placed afterward exhibit comparable robustness and smoothness to those with FFL placed beforehand.
>
> |Noise $\sigma$ | Metrics | FlipNet (FFL placed beforehand) | FlipNet (FFL placed afterward) |
> |---|---|---|---|
> | 0.05 | TAR |5503 ± 21 | 5499 ± 19 |
> ||AFR| 0.67 ± 0.02| 0.67 ± 0.02 |
> | 0.1 | TAR |5505 ± 13 | 5501 ± 14 |
> ||AFR|0.72 ± 0.02 |0.74 ± 0.01 |
>
> #### **> Question 5**
> Using complex matrices in the Fourier Filter Layer allows for **control over both the amplitude and the phase of frequency components**.
> After the Fourier transform, each element in $X$ is a complex number, i.e. $X_{u,v}=a+bj$.
> The corresponding frequency amplitude is determined by its magnitude $\sqrt{a^2+b^2}$,
> and the corresponding frequency phase is determined by its phase $\arctan\left(\frac{b}{a}\right)$.
> Similarly, after multiplying by the filter matrix $H$, the filtered frequency amplitude and phase are determined by the magnitude and phase of Hadamard product $X\odot H$, respectively.
>
> (1) When $H$ is a real matrix, the Hadamard product $X\odot H$ cannot change the frequency phases.
> By adjusting the element values in $H$, only the amplitude of each frequency can be altered (suppressed or enhanced).
> In this case, the output obtained by the inverse Fourier transform retains a clear physical meaning.
>
> (2) When $H$ is a complex matrix, the Hadamard product $X\odot H$ can change both the frequency amplitudes and the frequency phases.
> In this case, the sequence obtained by the inverse Fourier transform is not merely a linear combination of frequencies, but rather an arbitrary combination of frequency functions with arbitrary phases. This makes the output **more abstract, essentially becoming a type of feature.**
>
> Previously, the use of complex matrices in FFT has already been recognized as a feature extractor in NLP field$^{[1]}$.
>
> #### **> Question 6**
> 10 seeds are used in these figure and tables. The number of seeds has been specified in Appendix J now.
>
> #### **> Question 7**
> Different observation dimensions have different amplitudes. Based on the magnitude of the observations, we selected approximately 10% to 30% of the values as the noise magnitude.
>
> #### **> Question 8**
> Fig. 10 presents the action curve from a single experiment in a real-world driving task; hence, there is no error bar in it.
>
> Fig. 13 shows the average metrics of four scenarios from Fig. 28, rather than the average across multiple seeds, so there is no error bar in it.
>
> ---
> #### **Summary**
>
> We hope our response can address your concerns, and be glad to hear any remaining concerns.
> Given the unusual score from one reviewer,
> **we would be extremely grateful if you could consider increasing your score.**
> Looking forward to your response and appreciating your consideration!
>
> *References*
>
> *[1] J. Lee-Thorp, et al. FNet: Mixing Tokens with Fourier Transforms, ACL 2022.*

---

> > ### Comment · Reviewer_du6n · 2024-11-27
> >
> > I appreciate the authors' detailed responses. Some of my questions and concerns are addressed. I will increase my rating accordingly.
> >
> > ---
> >
> > For the authors' response to my question in the original review:
> > > In Equation 5, is the second term at the right hand side of the equation also determined by transition dynamics?
> >
> > I referred to whether $\xi_t$ could depend on $s_t$ in some problems or situations. More broadly, one thing I want to mention is that, for $o_t = s_t + \xi_t$, I don't think it is necessary the case in all possible situations, and again, whether $\xi_t$ could depend on $s_t$ is another thing.
> >
> > &nbsp;
> >
> > Another remaining concern of mine is that the noise models used in this work are basically uniform distributions (and maybe with manually decided ranges). A further study on the noise models/patterns supported by the evidence in practical problems will largely strengthen the paper, although this seems to be beyond the scope of this work a bit.
> >
> > But I think this does not affect the idea of using the Fourier filter layer in this paper. I do not see a big issue here at the moment.

---

> > > ### Author Response · Authors · 2024-12-01
> > > **Author Response**
> > >
> > > We sincerely thank you for your response.
> > >
> > > One of the contributions is identifying two fundamental reasons of action fluctuation shown in Equation (4),
> > > which provides a valuable insight for the community.
> > > The exact relationship between $\xi_t$ and $s_t$ you mentioned will be explored in future work.
> > > While noise models/patterns are not critical for Fourier filter layer (since it learns from the data whether each frequency should be suppressed/enhanced), we will also investigate various types of noise in future studies.
> > >
> > > Thank you again for the suggestions.

---

### Official Review · Reviewer_sQmH · 2024-11-02

**Soundness:** 1
**Presentation:** 1
**Contribution:** 1
**Rating:** 1
**Confidence:** 4

**Summary:**

This paper focuses on action fluctuation ratio and tries to minimize the consecutive action change between two temporally adjacent states. The submission claims to prove the Jacobian norm is an approximation of Lipschitz constant and proposes to add a Fourier filter layer to deal with the action change when there is observation noise. Some experiments in DeepMind Control Suite are provided. However, most of the experiments are conducted in a non-standard environment called “Double Integrator” and in a non-standard robotic application.

**Strengths:**

Enhancements in reinforcement learning would be of interest to the machine learning community.

**Weaknesses:**

Authors write in the submission in page 2 that:

*“The code is publicly released to facilitate the implementation and future research.”*

Again the authors write that at page 6

*“The code is publicly available at.”*

These are false statements. In the link the authors provide there is a GitHub repository and it states that:

*“Code and full article will be released after review stage.”*

If the authors did not want to share their code, they should honestly state in their paper as well that the code is not going to be shared.


The main theoretical contribution of the submission is not stated rigorously, you cannot just state that X is an approximation of Y in a formal theorem statement. Furthermore, the relationship between the norm of the Jacobian and the Lipschitz constant is a well-known fact that can be found in any multivariable Calculus textbook. The authors cannot claim that these results are their contributions.

The submission states as a main contribution that they identify the two fundamental reasons that cause action fluctuation. However, this is rather something obvious and has been clearly discussed before. Authors keep mentioning throughout the paper in the same manner as these “fundamental reasons”. I find this kind of writing quite misleading and unnecessary.

One problem I have is the action oscillation ratio metric itself that is introduced in [1]. It is not clear at all why we should measure this or try to minimize this metric. In high dimensional action and state space there can be differences in actions even though somehow the $\ell_p$-norm distance between two consecutive states is smaller does not mean the $\ell_p$-norm distance between the actions between these two states must also be smaller. It is not at all clear that this is a straightforward good approach. As far as I can see this action oscillation only appears in two papers in the past 5 years. I am not sure the submission's claims on how much researchers want to solve this extremely important point is accurate.

Furthermore, the action fluctuation ratio in Equation 4 that the submission states is a different metric than what the original paper states as action fluctuation ratio [1]. This should be also clearly stated here.

[1] Chen Chen, Hongyao Tang, Jianye Hao, Wulong Liu, Zhaopeng Meng. Addressing Action Oscillations through Learning Policy Inertia, AAAI 2021.


In section 4.2 in the Deepmind Control Suite results why is only the TD3 algorithm tested? There should be more baselines in the main training environment. SAC, TRPO and PPO should also be included in these results. Furthermore, in the DeepMind Control Suite it is usually tested in more environments including Hopper and Ant.

The caption for the bar graph in Figure 9 should not be covering the actual results. The results in the DeepMind Control Suite demonstrates that the total average return obtained by the FlipNet is within one standard deviation with the total average return obtained by MLP. From these results it is not clear FLipNet performs better. In particular, the total average return in the noise free Cheetah environment of MLP is 816$\pm$30 and 829$\pm$15 of FlipNet. The total average return in the noise free Reacher environment of MLP 981$\pm$10 and 988$\pm$10 for FLipNet. All of these results throughout the entire table are within one standard deviation.

It is also insufficient to only compare one method [1] in the main results.

[1]  Xujie Song, Jingliang Duan, Wenxuan Wang, Shengbo Eben Li, Chen Chen, Bo Cheng, Bo Zhang, Junqing Wei, Xiaoming Simon Wang. LipsNet: A Smooth and Robust Neural Network with Adaptive Lipschitz Constant for High Accuracy Optimal Control, ICML 2023.

**Questions:**

See above.

---

### Official Review · Reviewer_KBn3 · 2024-11-03

**Soundness:** 3
**Presentation:** 3
**Contribution:** 3
**Rating:** 6
**Confidence:** 4

**Summary:**

The paper investigates the issue of action fluctuations encountered in policy training for reinforcement learning, explicitly stating that this problem arises from two factors: policy non-smoothness and observation noise, and addresses these two causes in a decoupled manner. For policy non-smoothness, a Jacobian regularization technique is introduced to enhance the smoothness of the policy network. To address observation noise, a Fourier filter layer is incorporated to improve noise robustness. This paper is the first to decouple the study of the causes leading to action fluctuations, providing new insights for further research on this issue.

**Strengths:**

1. This paper clearly identifies the two fundamental causes of action fluctuations and resolves them through a decoupled approach.
2. This paper conducts rigorous formula derivations and proofs.
3. This paper implements good encapsulation of the network, facilitating further research and use.
4. This paper conducts experiments in various simulation and real-world environments, achieving state-of-the-art performance and promoting the application of reinforcement learning in the real world.

**Weaknesses:**

1. The compared methods should be consistent across different experimental environments. For example, methods (MLP-SN,    LipNet-G) should be compared in double integrator environment.
2. The figure in Section 3.4 lacks the figure number and caption.

**Questions:**

1. The compared methods should be consistent across different experimental environments.
2. In Equation (5), the second term $\frac{do_t}{d_t}$ reflects the change rate of $o_t$.  This change rate is not equal to level of observation noise. Why does this term reflect the level of observation noise?

---

> ### Author Response · Authors · 2024-11-19
> **Rebuttal by Authors**
>
> We thank you for the careful reading and insightful review!
>
> After carefully considering your concerns, we now address each of them in turn:
>
> ---
> #### **> Weakness 1 & Question 1**
> Sorry that you might be misled by our paper writing. In fact, MLP-SN and LipsNet-G **have already been compared in our original paper** using the double integrator environment. We kindly invite you to refer to **Fig. 8, Tab. 3, and Tab. 4**, which contain detailed comparison data. Also, the comparison results have already been discussed in **Line 378 ~ Line 384**.
>
> As for the DMControl environments, we have already compared MLP, LipsNet, and FlipNet.
> Also, the MLP-SN is compared on DMControl's reacher environment, whose result can be found in Tab. 13 in our original paper.
> What we want to emphasize is that compare MLP-SN on all DMControl environments is not realistic.
> Because this would necessitate the manual tuning of spectral norm hyperparameters for
> each layer in MLP-SN, which have an unwieldy number of potential hyperparameter combinations, as said in Appendix J.
>
> #### **> Weakness 2**
> Thank you for pointing out. We have added figure number and caption in Section 3.4, and kindly invite you to check it in the uploaded revision paper.
> #### **> Question 2**
> In RL control tasks, action is computed as $a_t = \pi(o_t)$ and observation is given by $o_t = s_t + \xi_t$, where $s_t$ represents the current state and $\xi_t$ denotes any type of observation noise.
> The term $\Vert \frac{\mathrm{d}o_t}{\mathrm{d}t} \Vert$ you mentioned is actually $\Vert \frac{\mathrm{d}o_t}{\mathrm{d}t} \Vert = \Vert \frac{\mathrm{d} (s_t + \xi_t)}{\mathrm{d}t} \Vert = \Vert \frac{\mathrm{d} s_t}{\mathrm{d}t} + \frac{\mathrm{d} \xi_t}{\mathrm{d}t} \Vert$. The first part $\frac{\mathrm{d} s_t}{\mathrm{d}t}$, i.e. the change rate of dynamics system, is related to the **inherent property** of the target dynamic system. Therefore, **when high-frequency noise occurs, the second part $\frac{\mathrm{d} \xi_t}{\mathrm{d}t}$ will causes $\Vert \frac{\mathrm{d}o_t}{\mathrm{d}t} \Vert$ to increase significantly.**
> Thank you for pointing out the potential source of misunderstanding. In the revised paper, we have decouple $o_t$ as two terms, i.e. $s_t$ and $\xi_t$, to make it clear. We kindly invite you to check Section 3.1 in the uploaded revision paper.
>
> ---
> #### **Summary**
>
> We hope our response can address your concerns, and be glad to hear any remaining concerns.
> Given the unusual score from one reviewer,
> **we would be extremely grateful if you could consider increasing your score.**
> Looking forward to your response and appreciating your consideration!

---

### Official Review · Reviewer_6JiW · 2024-11-04

**Soundness:** 2
**Presentation:** 4
**Contribution:** 4
**Rating:** 3
**Confidence:** 5

**Summary:**

This paper introduces a novel neural network architecture designed to address action fluctuation in reinforcement learning tasks. The proposed approach decomposes the problem into two distinct components: observational noise and policy smoothness. These challenges are addressed independently through a Fourier-based filter layer and a Lipschitz constant regularizer on the policy, respectively. Experimental results demonstrate that their method achieves superior performance compared to MLP-SN and Lipsnet across multiple tasks.

**Strengths:**

1. The theoretically grounded separation of observational noise and policy smoothness as distinct challenges to solve when approaching action fluctuation provides valuable insights for the community.
2. The paper includes clear and concise plots
3. The result that the jacobian norm can be used as a lipschitz constant estimate is an important discovery, and the proofs are rigorous
4. The sensitivity analysis is thorough and convincing, substantiating the claim that unlike some existing methods, their method is not significantly sensitive to the choice of hyperparameters.
5. The experimental section is thorough, including a real world test, and ablation studies, and comparisons against existing methods on standard benchmarks.

**Weaknesses:**

1. The authors present that one their main contributions is that Flipnet can be used just like an mlp in standard RL algorithms, making it seem like a drop in replacement for existing RL algorithm implementations based on pytorch. However, this does not seem possible to me as the fourier transform layer specifically requires the history of observations to be stored. Most standard implementations do not store the history of observations as they assume that the environment is a markov decision process, and often rely on this assumption to make their implementations efficient.
2. The code is also defined as publicly available but it is not provided until after the review process is complete, which means I can't verify their previous claim.
3. The backwards pass is approximately 3x slower than Lipsnet and 10x slower than a standard MLP, and forward computation is twice as expensive as MLP-SN.
4. Lack of any mention of the limitations of the method in the main paper (for example comutational time is burried in the appendix). They dismiss alternative methods as needing substantial changes to existing RL algorithms, thereby not needing to compare against them, but many of these methods can also be cast as part of the neural network architecture, for example CAPS and L2C2 can be defined in the backwards pass as part of their loss functions.

If the authors resolve the limitations representation issue presented above, I will substantially improve my recommendation for the acceptance of this paper and my soundness score. It is otherwise strong enough to be publishable regardless of the performance against the other related methods and computational time concerns.

**Questions:**

1. Have the authors considered using more efficient filtering methods, such as Butterworth filters, to reduce computational overhead, since the motivation is real time applications?
2. How does the architecture perform in environments requiring rapid reaction times?
3. Can the authors provide more justification for using complex matrices in feature extraction? As it seems to go against the notion of separation of concerns.

---

> ### Author Response · Authors · 2024-11-19
> **Rebuttal by Authors (1/2)**
>
> We thank you for the careful reading and insightful review!
>
> After carefully considering your concerns, we now address each of them in turn:
>
> ---
> #### **> Weakness 1**
> **Using historical observations is a very common practice** for the following three reasons:
>
> (1). You can easily find *ReplayBuffer* classes for storing historical observations in various RL training tools.
> For instance, the *num_steps* parameter in *TensorFlow Agents* ([link](https://tensorflow.google.cn/agents/api_docs/python/tf_agents/replay_buffers/replay_buffer/ReplayBuffer?hl=en)), the *stack_num* parameter in *Tianshou* ([link](https://tianshou.org/en/stable/01_tutorials/07_cheatsheet.html)), and the *sequence* parameter in *RLlib* ([link](https://docs.ray.io/en/latest/rllib/package_ref/doc/ray.rllib.utils.replay_buffers.replay_buffer.StorageUnit.html#ray.rllib.utils.replay_buffers.replay_buffer.StorageUnit)) are all for historical observations usage.
>
> (2). In POMDP$^{[1]}$ or RL algorithms like Dreamer$^{[2]}$, they all use historical observations as a very common setting.
>
> (3). In real-world control tasks, using historical observations is essential—for example, in autonomous driving, this has become a widely accepted consensus.
>
> #### **> Weakness 2**
> Thank you for your feedback. Now, the code is accessible in our anoymous GitHub ([link](https://github.com/ICLR-anonymous-2025/FlipNet)).
>
> #### **> Weakness 3**
> (1). For the forward time: As shown in Tab. 6, FlipNet's 1-batchsize forward time is 0.16ms, and that of MLP is 0.10ms.
> Although 0.16ms appears to be 2x slower than 0.10ms, **the absolute value of 0.16ms is very small and fully meets the requirements for real-time inference.**
>
> (2). For the backward time: 10x backward time does not mean the training time is 10x slower. Backward propagation is just one step in RL. RL involves additional time-consuming steps beyond backward, such as sampling and evaluation. To more accurately illustrate the impact on training time, we list the training wall-clock times for 1M iterations of TD3 on DMControl envs below.
> | Network  | Cartpole | Reacher | Cheetah | Walker | Total  |
> |----------|----------|---------|---------|--------|--------|
> | MLP  | 120 min  | 118 min | 128 min | 125 min| 491 min|
> | FlipNet | 194 min  | 195 min | 206 min | 204 min| 799 min|
>
> On average, the training time of FlipNet is 1.6 times that of MLP, which more accurately illustrates the impact on training time. The result is updated in Appendix H. We kindly invite you to check it in the uploaded revision paper.
>
> #### **> Weakness 4**
> Besides not being network-level methods, we did not evaluate CAPS and L2C2 also due to their high sensitivity to penalty coefficients.
> Tuning these hyper-parameters to balance their performance and smoothness for all tasks in our paper is not practical.
> However, in order to address your concern, we have added comparative experiments with CAPS and L2C2 by carefully tuning in the double integrator environment.
>
> Total average return comparison:
> - |Noise $\sigma$ | MLP | CAPS | L2C2 | MLP-SN | LipsNet-G | LipsNet-L | FlipNet |
> |---|---|---|---|---|---|---|---|
> | 0.01 | **-51.0**±0.1 | -52.1±0.1 | -55.0±0.1 | -62.0±0.1 | -53.2±0.1 | -55.3±0.1 | -56.5±0.1 |
> | 0.05 | -53.5±0.2 | **-53.0**±0.2 | -55.4±0.4 | -62.3±0.4 | -54.3±0.3 | -55.6±0.4 | -56.8±0.2 |
> | 0.1 | -59.5±0.6 | -56.9±0.6 | -57.5±0.5 | -62.8±0.7 | **-54.2**±0.7 | -56.0±0.6 | -57.1±0.6 |
> | 0.2 | -78.4±1.8 | -67.9±1.0 | -63.4±1.5 | -65.9±1.7 | -59.8±1.1 | -58.7±1.4 | **-57.9**±0.6 |
> | 0.3 | -103.2±3.7 | -85.9±3.0 | -82.3±4.4 | -71.8±2.3 | -74.3±2.1 | -65.3±1.6 | **-59.9**±1.9 |
>
> Action fluctuation ratio comparison:
> - |Noise $\sigma$ | MLP | CAPS | L2C2 | MLP-SN | LipsNet-G | LipsNet-L | FlipNet |
> |---|---|---|---|---|---|---|---|
> | 0.01 | 0.02±0.01 | 0.01±0.01 | 0.01±0.01 | 0.01±0.01 | 0.01±0.01 | 0.01±0.01 | **0.00**±0.01 |
> | 0.05 | 0.11±0.01 | 0.07±0.01 | 0.05±0.01 | 0.03±0.01 | 0.04±0.01 | 0.03±0.01 | **0.01**±0.01 |
> | 0.1  | 0.19±0.01 | 0.15±0.01 | 0.10±0.01 | 0.06±0.01 | 0.07±0.01 | 0.06±0.01 | **0.02**±0.01 |
> | 0.2  | 0.34±0.02 | 0.26±0.01 | 0.20±0.01 | 0.13±0.01 | 0.17±0.01 | 0.12±0.01 | **0.03**±0.01 |
> | 0.3  | 0.48±0.02 | 0.37±0.02 | 0.33±0.02 | 0.20±0.01 | 0.28±0.01 | 0.20±0.01 | **0.05**±0.01 |
>
> The results show that CAPS and L2C2 cannot balance performance and smoothness well, and FlipNet is still the SOTA method.
> We kindly invite you to check Tab. 3, Tab. 4, and Fig. 8 in the uploaded revision paper, where the new results have been added.
>
> Continue in the next reponse ...

---

> ### Author Response · Authors · 2024-11-19
> **Rebuttal by Authors (2/2)**
>
> Continue here ...
>
> #### **> Question 1**
> Yes, we have considered it. However, although the Butterworth filter has lower computational overhead, we aim to **avoid introducing a predefined response curve shape**, which could significantly degrade control performance.
> In contrast, our Fourier Filter Layer learns from the data whether each frequency should be suppressed or enhanced, offering higher flexibility without sacrificing performance.
> If the data happens to be suitable, our Fourier Filter Layer can learn a Butterworth filter.
> Additionally, its forward time is within 0.2 ms, meeting the requirements for real-time processing.
>
> #### **> Question 2**
> FlipNet performs well in rapid reaction environments.
> The DMControl-Cheetah is such a environment,
> where the actions need to change rapidly between -1 and 1 to make cheetah move fast.
> Some figures and data are provided in [LINK](https://github.com/ICLR-anonymous-2025/FlipNet/blob/master/rebuttal/For_reviewer_%236JiW.md) to illustrate its effectiveness.
>
> #### **> Question 3**
> Using complex matrices in the Fourier Filter Layer allows for **control over both the amplitude and the phase of frequency components**.
> After the Fourier transform, each element in $X$ is a complex number, i.e. $X_{u,v}=a+bj$.
> The corresponding frequency amplitude is determined by its magnitude $\sqrt{a^2+b^2}$,
> and the corresponding frequency phase is determined by its phase $\arctan\left(\frac{b}{a}\right)$.
> Similarly, after multiplying by the filter matrix $H$, the filtered frequency amplitude and phase are determined by the magnitude and phase of Hadamard product $X\odot H$, respectively.
>
> (1) When $H$ is a real matrix, the Hadamard product $X\odot H$ cannot change the frequency phases.
> By adjusting the element values in $H$, only the amplitude of each frequency can be altered (suppressed or enhanced).
> In this case, the output obtained by the inverse Fourier transform retains a clear physical meaning.
>
> (2) When $H$ is a complex matrix, the Hadamard product $X\odot H$ can change both the frequency amplitudes and the frequency phases.
> In this case, the sequence obtained by the inverse Fourier transform is not merely a linear combination of frequencies, but rather an arbitrary combination of frequency functions with arbitrary phases. This makes the output **more abstract, essentially becoming a type of feature.**
>
> Previously, the use of complex matrices in FFT has already been recognized as a feature extractor in NLP field$^{[3]}$.
>
> ---
> #### **Summary**
>
> We hope our response can address your concerns, and be glad to hear any remaining concerns.
> Given the unusual score from one reviewer,
> **we would be extremely grateful if you could consider increasing your score.**
> Looking forward to your response and appreciating your consideration!
>
> *References*
>
> *[1] G. Shani, et al. A Survey of Point-based POMDP Solvers, AAMAS, 27, 1-51, 2013.*
>
> *[2] D. Hafner, et al. Mastering Atari with Discrete World Models, ICLR 2021.*
>
> *[3] J. Lee-Thorp, et al. FNet: Mixing Tokens with Fourier Transforms, ACL 2022.*

---

> > ### Comment · Reviewer_6JiW · 2024-11-23
> > **Response to rebuttal**
> >
> > Weakness 1. I am aware that using historical observations is available in some RL libraries, but it is not available in others (custom implementations, cleanrl, spinup, etc...), the reason why this is a weakness is because it is touted as a main contribution that this network level method can be just placed instead of an MLP. This is not true in many workloads, for example implementations on real robots may assume MDP processes (I have such workloads). The fact that there are at least some cases where changes are necessary for the RL algorithm nullifies the argument that it does not require changes in existing RL algorithms. This is a limitation of the method, whether it's a problem or not depends on the RL algorithm implementation. To give constructive feedback, I believe a limitations section would strengthen the paper, even if after the individual limitations an argument is given for why it is not a large limitation (such as that it is common for historical observations to be available). I would also decrease the focus on the contribution that represents it as a drop-in replacement for MLPs.
> >
> > Weakness 2. Code being presented is appreciated!
> >
> > Weakness 3. This touches on the same issues I have with weakness 1, the issue is not the limitation itself, the issue is that the limitations are not presented. I understand that in your setting the NN runs fast enough for real time use regardless of extra cost, but if I run this on an RP2040, suddenly the difference does matter. The extra cost is there and should be presented as so, not doing so is misleading.
> >
> > Weakness 4. Appreciate the comparison against the methods I gave as an example, and they actually strengthen the paper in my opinion, because now you don't have to make the argument of one being a network level method and the other not. However, the first sentence of weakness 4 is "Lack of any mention of the limitations of the method in the main paper", this was ignored, and is by far my primary concern.
> >
> > Questions: Your responses to the questions are excellent, thank you for the thorough extrapolations.
> >
> > I have a new concern arising from one of the existing reviews (namely the one by sQmH), regarding the relationship between the Lipschitz constant and the Jacobian norm. The paper has a major contribution presenting the Jacobian norm as an approximation of the local Lipschitz constant. How does this relate to existing results relating the Lipschitz constant and the Jacobian norm (for example https://encyclopediaofmath.org/wiki/Lipschitz_constant)? I noticed that there are some differences but I would like a clearer picture around how it fits in with existing mathematical results.

---

> ### Author Response · Authors · 2024-11-24
> **Rebuttal by Authors**
>
> Thank you for your feedback.
>
> #### **> Weakness 1**
> We respect your workloads on real robots, where using history observations is widely recognized as essential. While it's possible to list RL libraries that do or do not support historical observations, this is beyond the scope of our paper; our focus is on facilitating smoother control for your real robot scenarios.
>
> Furthermore, the support of history observation can be achieved with minimal effort—using **a single line of code** to call a wrapper `gymnasium.wrappers.FrameStackObservation` ([link](https://gymnasium.farama.org/api/wrappers/observation_wrappers/#gymnasium.wrappers.FrameStackObservation)), applicable to any RL library. This approach also preserves the Markov property, as the augmented state $[s_t, s_{t-1}, ..., s_{t-N}]$ forms a new MDP. Given your workloads with real robots, we trust you may familiar with this straightforward solution.
>
> We don’t focus on which RL library you use or how you implement the wrapper; **replacing the MLP with FlipNet in tasks with historical observations achieves smoother control—this is our straightforward contribution.**
>
> #### **> Weakness 3 & 4**
> Thank you for suggesting a Limitation Section. We have added Appendix N for this purpose.
>
> #### **> New concern**
> 1. The reference link focuses on the global Lipschitz constant, while our **Theorem 3.1 emphasizes the local Lipschitz constant**. Neural networks often exhibit varying behaviors across regions, whose global Lipschitz constant dominated by steep-gradient areas. This makes it less representative and meaningful for most input points, especially in complex machine learning tasks. Our theorem fills this gap by offering a localized analysis.
> 2. Unlike the reference link, which provides an upper bound equaling the Lipschitz constant **only in convex domains**, our Theorem 3.1 imposes **no such assumptions** and instead establishes that the Jacobian norm serves as **a first-order approximation** of the local Lipschitz constant.
> 3. Most importantly, prior works like MLP-SN and LipsNet directly optimize the Lipschitz constant—a challenging approach. In Theorem 3.1, the relationship between local Lipschitz constant and Jacobian norm **shows that regularizing Jacobian norm is a simpler way to smooth policy networks, offering valuable insight for the community.**
>
> We sincerely appreciate your active feedback and specific suggestions.

---

> ### Comment · Reviewer_6JiW · 2024-11-25
> **Final response**
>
> It is clear that the authors are not taking my negative suggestions seriously, the limitations section they added is buried as the last section of the appendix, and does not represent the limitations in any meaningful manner. Instead, the section is now about future work more than it is about limitations, and the limitations themselves are heavily downplayed. No, the FrameStackObservation wrapper doesn't solve the problem. Adding history to the observational space means adding it to the inputs of the Value functions that these RL algorithms use, which is not what this method intends to do, it also wastes memory as the observations are duplicated for each state to keep it an MDP. Even if I accept the wrapper argument, this allows for having historical observations during training, not during inference where the constraints of implementations apply. It's clear the authors will only give arguments against my concerns rather than incorporate them improving the paper. Even though they have addressed some of my other concerns, my main issue was with limitations representation, as this is still far from improved in the paper, I will lower my score.

---

> > ### Author Response · Authors · 2024-11-25
> > **Rebuttal by Authors**
> >
> > #### **> Limitation section**
> > Although the training time limitation has already been detailed in Appendix H, we have nonetheless added a Limitation section as per the reviewer’s suggestion.
> > In the Limitation section (Appendix N), we not only describe the limitation at the beginning sentence of the paragraph,
> > but also provide direct solution to address it—this corresponds to the "future works" mentioned by the reviewer.
> > We do not believe this presents any issues.
> >
> > #### **> Wrapper**
> > The reviewer said the wrapper `gymnasium.wrappers.FrameStackObservation` ([link](https://gymnasium.farama.org/api/wrappers/observation_wrappers/#gymnasium.wrappers.FrameStackObservation)) does not work because inputs of Value functions are augmented.
> > We would like to remind the reviewer that Python has a slicing feature.
> > Your issue can be resolved with a single line of code: `s_v = s[-1, :]`, where `s_v` is the input for V, and `s` is the observation from the wrapper.
> >
> > The reviewer said the wrapper only works for training but not for implementation,
> > but this is an engineering implementation issue in real-world tasks, out of scope for our paper.
> > Additionally, we have already conducted experiments on real-world mini-vehicles (Section 4.3), and the engineering implementation of stacking observations is very simple.
> >
> > ---
> > Thank you for your feedback. Let us emphasize once again: What device and RL library you use, or how you implement the wrapper, are just minor details. **Replacing the MLP with FlipNet in tasks with historical observations achieves smoother control—this is our straightforward contribution.**

---

### Meta-Review · Area_Chair_FxDB · 2024-12-24

**Metareview:**

Smooth actions are critical to handle in tasks like robotics and other control problem. This paper addresses the action fluctuation problem in RL, which is a phenomenon where small differences in adjacent states can yield large differences in consecutively generated actions. The authors argue that two fundamental factors underlie these fluctuations - the non-smoothness of the policy network and noise in observations. They then propose a novel policy architecture that integrates two core techniques to improve policy smoothness via jacobian regularization and improved robustness to noisy inputs via a fourier filter layer.  Flipnet can be dropped into actor critical methods like TD3/DSAC. Results are shown on DM Control suite, a linear quadratic control task and a mini vehicle driving test.

Strengths: Authors show via local analysis that jacobian norm acts as a good proxy for a networks local lipschitz constant + introduction of a trainable fourier filter layer. This is empirically shown to work better on synthetic and real tasks requiring smooth actions. This paper's strength mainly relies on its experimental results, and therefore they need to be stronger to investigate the extent of the contribution. Although the authors responded by adding results in environments like Hopper, Ant, and Humanoid, there was feedback suggesting broader or more diverse tasks might give further confidence in FlipNet’s benefits.

Smooth actions are mostly critical in real world physical problems. But then one can take a more principled approach to take model based and model free methods to handle actions in a more robust way. I think the authors should have a physical robot test and compare with standard industry baselines. Without this, it will be hard to assess the evidence of real world impact. It is not like the empirical performance is better on other dimensions - it is mostly shown to improve action smoothness, which is significant but warrants stronger real world experimentation. I think this is why the paper did not get higher scores.

**Additional Comments On Reviewer Discussion:**

Overall this is an interesting paper and it studies an important problem for real world applications of reinforcement learning. Although the rebuttal process addressed many of the concerns, I think there was lack of consensus amongst the reviewers to give better scores for a clear acceptance. I would encourage the authors to take all of this feedback into consideration and resubmit at a later stage.

---

### Decision · Program_Chairs · 2025-01-22

Reject